# HF-FONT: FEW-SHOT FONT GENERATION VIA HIGH-FREQUENCY STYLE ENHANCEMENT AND FUSION

## ABSTRACT

Few-shot font generation aims to create new fonts with a limited number of glyph references. It can be used to greatly reduce the workload of manual font design. However, although existing methods have achieved satisfactory performance, they still struggle to capture delicate glyph details, thus resulting in stroke errors, artifacts, and blurriness. To address these problems, we propose HF-Font, a novel framework that generates fonts with higher structural fidelity. Specifically, inspired by the observation that high-frequency information of character images often contains distinct style patterns (*e.g.*, glyph topology and stroke variation), we develop a novel style-enhanced module to improve the style extraction by incorporating high-frequency features from reference images using a high-pass filter. Then, for guiding the generation process, we design a Style-Content Fusion Module (SCFM), which integrates the style features with a component-wise codebook that encodes content semantics. Moreover, we also introduce a style contrastive loss to better transfer high-frequency features. Extensive experiments show that our HF-Font outperforms the state-of-the-art methods in both qualitative and quantitative evaluations, demonstrating its effectiveness across diverse font styles and characters. Our source code will be released soon.

## 1 INTRODUCTION

The task of few-shot font generation allows to transfer the font style from a source domain to a target domain based on only a few reference images. It can greatly alleviate the burden of time-consuming and labor-intensive manual design, especially for some character-rich languages like Chinese, Japanese, or Korean. Therefore, few-shot font generation techniques can benefit many critical applications, including logo design, ancient character restoration, and so on.

At present, with the rapid development of deep learning architectures, such as Convolutional Neural Networks (CNNs) (Chen et al., 2022), Generative Adversarial Networks (GANs) (Goodfellow et al., 2014), and diffusion models (Ho et al., 2020), researchers have made great advances in creating gratifying fonts. A widely-adopted strategy for this task is the style-content disentanglement paradigm, which decouples the content and style representations from the given content and reference images. These two representations are then combined and decoded to generate the target glyph. Early approaches (Zhang et al., 2018b; Gao et al., 2020; Xie et al., 2021; Wang et al., 2023) mainly model font style as a set of universal statistic features, and utilize the extracted representations to encode stylistic information. In addition, witnessing the structure variations and local correlations in font styles, some works (Park et al., 2021b; Tang et al., 2022; Kong et al., 2022; Chen et al., 2024) also employ structure-aware representations, which typically decompose characters into different components and acquire multiple style representations to boost the performance. Nevertheless, despite the remarkable progress, existing methods still have several drawbacks. The highly diverse and intricate nature of font styles often leads to obvious defects in synthesized results, like incomplete or unwanted strokes, anomalous blurriness, and artifacts. Besides, a core challenge of few-shot font generation lies in how to accurately capture the target style from just a few reference images. Previous works tend to exhibit a limited performance in reproducing stylistic details, due to their insufficient style extraction ability.

To tackle the above challenges, our key idea revolves around leveraging high-frequency features to enhance the style extraction. As shown in Figure 1, these high-frequency components are defined by

Figure 1: **Some character samples and their corresponding high-frequency components.** We find that high-frequency components provide clearer contours and highlight distinctive style patterns. Zoom in to see details.

rapid spatial intensity transitions along stroke boundaries, rather than by differences in color. They highlight the overall contours of the glyphs and reveal essential style attributes such as topology, stroke variation, and line spacing. Hence, the incorporation of high-frequency information facilitates a more effective and robust style extraction process.

In light of this, we put forward a new end-to-end few-shot font generation method called HF-Font. To be specific, we first develop a style-enhanced module to process both style reference images and their corresponding high-frequency counterparts in parallel. Given that the reference images usually contain background noise, we introduce a gate mechanism to filter out the irrelevant information. Subsequently, the style features extracted from both branches are adaptively fused with the component-wise content representation through an innovative Style-Content Fusion Module (SCFM), which includes two Grouped Residual Attention Blocks (GRABs) and one output adapter. Apart from these, we also design a style contrastive loss to further obtain discriminative features, promoting both realism and diversity in the generated results. Consequently, our method yields superior and impressive stylization performance, highlighting its potential for practical applications.

To summarize, our major contributions are as follows:

- We propose HF-Font, a novel few-shot font generation framework. It incorporates a style-enhanced module to leverage high-frequency information from reference images, thereby strengthening structural integrity.

- We devise an inventive Style-Content Fusion Module (SCFM), which efficiently integrates style information with the component-level codebook. Furthermore, a style contrastive loss is also employed to facilitate the transfer of high-frequency features.

- Extensive experimental results verify that our HF-Font surpasses the leading models in both qualitative and quantitative evaluations, which demonstrates its effectiveness and generalization capability across diverse font styles and characters.

## 2 RELATED WORK

### 2.1 IMAGE-TO-IMAGE TRANSLATION

Image-to-image (I2I) translation is to convert an image from one domain to another while preserving its semantic content. Previous methods, such as Pix2pix (Isola et al., 2017) and CycleGAN (Zhu et al., 2017), mainly leverage GANs (Goodfellow et al., 2014) for supervised or unsupervised settings. Later, motivated by the human ability of inductive reasoning, FUNIT (Liu et al., 2019) employs Adaptive Instance Normalization (AdaIN) (Huang & Belongie, 2017) to fuse the encoded content and style features. Recently, diffusion models have emerged as a powerful generative technology. ILVR (Choi et al., 2021) achieves high-quality performance using only a pre-trained unconditional Denoising Diffusion Probabilistic Model (DDPM) (Ho et al., 2020). SCDM (Ko et al., 2024) applies stochastic perturbations to semantic maps and conditions I2I translation on the diffused labels. Notably, since font generation can be viewed as a special I2I translation task, many generic I2I translation approaches can be adaptively modified for font generation.

## 2.2 FEW-SHOT FONT GENERATION

Few-shot font generation intends to create a required font library with just a handful of glyph references. A classical tactic involves extracting content and style features from the input source and reference images, respectively. EMD (Zhang et al., 2018b) and AGIS-Net (Gao et al., 2020) disentangle the representations of style and content, and model each font as a universal descriptor. DG-Font (Xie et al., 2021) introduces a feature deformation skip connection to capture the glyph deformations. CF-Font (Wang et al., 2023) extends DG-Font through incorporating a content fusion module to narrow the gap between the source and target fonts. Later, NTF (Fu et al., 2023) formulates font generation as a continuous transformation via a neural transformation field (Mildenhall et al., 2020). FontDiffuser (Yang et al., 2024) implements a multi-scale content aggregation block along with a style contrastive refinement module to guide the whole framework.

For the generation of highly-structured characters, some notable studies leverage prior domain knowledge, such as stroke decomposition and trajectory, to optimize the final results. SA-VAE (Sun et al., 2018) stands out as the first attempt to integrate radicals and spatial structures into the generative model. LF-Font (Park et al., 2021b) represents component-wise style feature via a low-rank matrix factorization. MX-Font (Park et al., 2021a) utilizes a multi-head design, with each head weakly supervised to extract distinct local concepts. MX-Font++ (Wang et al., 2025) extends MX-Font by introducing Heterogeneous Aggregation Experts (HAE) to enhance feature extraction and decoupling. Afterwards, CG-GAN (Kong et al., 2022) utilizes a component predictor to assist the generator during adversarial training, while FsFont (Tang et al., 2022) establishes a character-reference mapping relationship and employs cross-attention to align the patch-level features. Diff-Font (He et al., 2024) infuses predefined embedding tokens into the condition diffusion model to support the sampling process. IF-Font (Chen et al., 2024) replaces the source image with the Ideographic Description Sequence (IDS) to control glyph semantics. Yet, these methods remain dependent on the labels of component categories. In contrast, VQ-Font (Pan et al., 2023) constructs a vector quantization-based encoder to automatically extract components. DA-Font (Chen et al., 2025) integrates a Dual-Attention Hybrid Module to optimize style transfer and relation-aware feature harmonization. Nevertheless, prevailing approaches still fall short in effectively encapsulating intricate structural features. This motivates us to harness high-frequency information and design a novel feature fusion strategy, as such information offers sharper style cues.

## 3 METHODOLOGY

### 3.1 OVERALL SCHEME

Given a set of $k$ reference images $x = \{x_i\}_{i=1}^k$ and a content image $I_c$, our model aims to generate a character $I_o$ that retains the same content with $I_c$ and the same style with $x$. As illustrated in Figure 2 (a), the generator mainly includes eight parts. Among them, the content encoder $E_c$ extracts the feature representation $f_c$ from $I_c$. It is pre-trained via a Vector Quantized Variational Auto-Encoder (VQ-VAE) to acquire a component-wise codebook $F_c$ as well. The reference images $x$ are first processed by a high-pass filter to obtain their high-frequency images $h = \{h_i\}_{i=1}^k$. Next, the spatial and high-frequency style encoders, $E_{ss}$ and $E_{sh}$, extract style features $f_{spa} = \{f_{is}\}_{i=1}^k$ and $f_{fre} = \{f_{ir}\}_{i=1}^k$ from $x$ and $h$, respectively. After that, $f_{spa}$ is further refined through a gate mechanism to yield $\hat{f}_{spa} = \{\hat{f}_{is}\}_{i=1}^k$, which is then fused with $f_{fre}$ and $F_c$ via our proposed SCFM to derive the style feature $f_{sa}$. Meanwhile, the Content Alignment Module (CAM) adaptively re-weights and aggregates $\hat{f}_{spa}$ to get the aligned content feature $f_{ca}$. Finally, the decoder $D_r$ outputs the result $I_o$ by concatenating $f_c$, $f_{sa}$, and $f_{ca}$ as inputs.

During the model's training, a multi-task discriminator $D_{s,c}$ is employed to distinguish between the real and generated images, as shown in Figure 2 (b). Moreover, to ensure that the generated glyph accurately reflects the reference style while preserving the input content structure, it also performs the classifications on each character, identifying both its style and content categories. The complete architectures of the generator and discriminator are provided in Appendix A.1. Notably, the content encoder $E_c$ and the codebook $F_c$ remain frozen during the main training stage, while all other components are fully trainable.

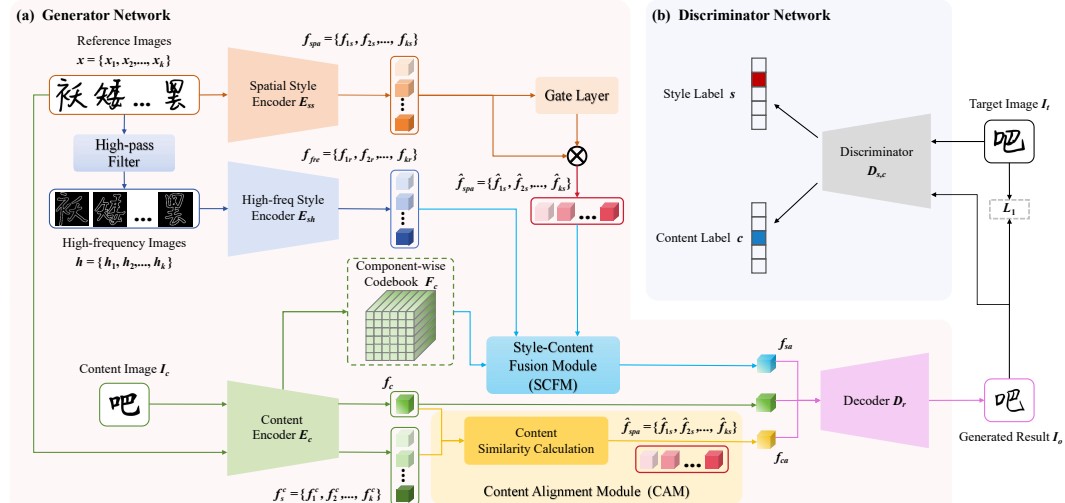

Figure 2: **Overview of our proposed HF-Font.** (a) The generator network mainly consists of the following parts: a pre-trained content encoder $E_c$, a spatial style encoder $E_{ss}$, a high-frequency style encoder $E_{sh}$, a high-pass filter, a gate mechanism, a Content Alignment Module (CAM), a Style-Content Fusion Module (SCFM), and a decoder $D_r$. (b) A discriminator network is used to distinguish the real and fake images, while also classifying the content and style categories of the generated characters.

## 3.2 STYLE-ENHANCED MODULE

To enrich the style representations, we introduce a style-enhanced module that exploits high-frequency information into the feature extraction process. In particular, a Laplacian kernel is used as a high-pass filter to obtain high-frequency counterparts $h$ from reference images $x$. This operation efficiently emphasizes fine-grained details without relying on any frequency-domain transformation (Wahl, 2024). Next, two parallel style encoders, $E_{ss}$ and $E_{sh}$, are employed to extract complementary style features $f_{spa}$ and $f_{fre}$ from $x$ and $h$ separately. Though structurally identical, these two encoders do not share weights with each other. Moreover, a gate mechanism is integrated into the spatial branch to modulate $f_{spa}$, allowing only informative style cues to pass.

**Gate Mechanism.** Intuitively, the stroke regions in reference images are often sparse, which makes style extraction susceptible to background noise (Dai et al., 2024). Here, background noise refers to the non-informative or low-response activations in intermediate feature maps during convolutional encoding, rather than the white background of the glyph images. To solve this problem, we bring in a gate mechanism to selectively filter the reference features. Specifically, the initial spatial style feature $f_{spa} = \{f_{is}\}_{i=1}^{k}$ is fed into a gate layer, consisting of a learnable fully-connected layer followed by a Sigmoid activation, to produce the gate units $w = \{w_i\}_{i=1}^{k}$. Each $w_i$ flexibly controls the information flow of $f_{is}$, where higher values indicate stronger retention of the style signals. The refined features $\hat{f}_{spa} = \{\hat{f}_{is}\}_{i=1}^{k}$ are obtained by element-wise multiplication, i.e., $\hat{f}_{is} = w_i \cdot f_{is}$. This design effectively suppresses extraneous noise while preserving meaningful style patterns.

## 3.3 GLYPH FEATURE DECOMPOSITION AND CONTENT ALIGNMENT MODULE

**Glyph Feature Decomposition.** In our HF-Font, the content encoder is pre-trained through a glyph feature decomposition network, which projects each character into a component-wise codebook. This network is trained on a certain character set for image reconstruction. As illustrated in Figure 3, the content encoder $E_c$ is built on CNN and maps a character image $I_f$ into latent representation $Z_c$. Then, vector quantization (Aaron et al., 2017) is applied to discrete $\hat{Z}_c$ as:

$$z_c^i = e_c^j, \quad \text{s.t.} \quad j = \underset{j \in \{1,2,\dots,d\}}{\arg\min} \|z_c^i - e_c^j\|_2^2 \tag{1}$$

where each spatial vector $z_c^i$ in $Z_c$ is replaced by its nearest code vector $e_c^j$ from the codebook $F_c$, which contains $d$ code vectors. Finally, the reconstruction decoder $D_r$ takes the matched codes $Z_q$ as input to reconstruct the glyph $I_r$. Visual analyses of the codebook are presented in Appendix A.2.

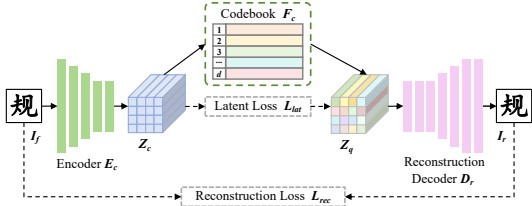

Figure 3: **The architecture of glyph feature decomposition network.** It is utilized for pre-training $E_c$ and acquiring $F_c$.

During pre-training, $E_c$ and $D_r$ are optimized by minimizing an objective function $\mathcal{L}_{pre}$ that contains a reconstruction loss $\mathcal{L}_{rec}$ and a latent loss $\mathcal{L}_{lat}$, defined as follows:

$$\mathcal{L}_{pre} = \mathcal{L}_{rec} + \mathcal{L}_{lat} = \|I_f - I_r\|_1 + \alpha\|\text{sg}[Z_c] - Z_q\|_2^2 + \gamma\|Z_c - \text{sg}[Z_q]\|_2^2 \tag{2}$$

where sg denotes the stop-gradient operator. $\alpha$ and $\gamma$ are the balancing hyper-parameters. Experimentally, we set them as 1 and 0.25, respectively.

Upon completing pre-training, we fix the content encoder $E_c$ along with the codebook $F_c$ to build the font generation model. To be noticed, although the reconstruction decoder $D_r$ adopts the same architecture with the font generation model's decoder, the latter is re-trained from scratch during the main training phase.

**Content Alignment Module.** From a perceptual perspective, reference characters sharing similar elements with the input glyph should receive more attention during the style transfer process (Zhu et al., 2020). To this end, in the Content Alignment Module (CAM), we first extract the content features $f_s^c = \{f_i^c\}_{i=1}^k$ from reference images via the content encoder $E_c$. In addition, to ensure dimension compatibility, both $f_s^c$ and $f_c$ are reshaped to $\overline{f_s^c}$ and $\overline{f_c}$. Then, the similarity values are calculated using the normalized cross-correlation measurement (Singhal, 2001) as:

$$\Phi_{ia} = \frac{\langle \overline{f_i^{ca}}, \overline{f_c^a} \rangle}{\|\overline{f_i^{ca}}\| * \|\overline{f_c^a}\|}, \quad a \in \{1, 2, \ldots, T\} \tag{3}$$

where $a$ refers to the position within the $T$-dimensional channel, and $\Phi_{ia}$ is a scalar representing the similarity between the $a$-th channel of the $i$-th reference image and the content image. Next, we normalize these values channel-wise using the Softmax function, and apply them to weight the spatial style feature representations $\hat{f}_{spa}$ to obtain the aligned content feature $f_{ca}$ by:

$$\overline{\Phi}_{ia} = \text{Softmax}(\Phi_{ia}) \qquad f_{ca} = \text{Concat}_a\left(\sum_{i=1}^{k} \overline{\Phi}_{ia} \hat{f}_{is}^a\right) \tag{4}$$

### 3.4 STYLE-CONTENT FUSION MODULE

The Style-Content Fusion Module (SCFM) serves as the core integration unit that harmonizes content structure and style representations. As shown in Figure 4, each attention block contains two key components, namely Grouped Residual Layer (GRL) and Exponential-Space Relative Position Bias (ES-RPB), which are introduced below.

**Grouped Residual Layer.** Some prior works (Zhou et al., 2023; Li et al., 2024) have pointed out the computational redundancy and complexity in the generation of query, key, and value matrices. To mitigate this, we design the Grouped Residual Layer (GRL) that integrates grouping and residual connections to improve efficiency and learning capacity. Formally, given an input $S \in \mathbb{R}^{B \times N \times C}$, we first split $S$ along the channel dimension into two halves, denoted as $S_1, S_2 \in \mathbb{R}^{B \times N \times \frac{C}{2}}$. Then, two linear layers, $\Gamma_1$ and $\Gamma_2$, are applied to both parts, combined with residual connections to enhance the gradient flow, as follows:

$$\begin{aligned} \Omega_1 &= \Gamma_1(S_1) + S_1, \quad \Omega_1 \in \mathbb{R}^{B \times N \times \frac{C}{2}} \\ \Omega_2 &= \Gamma_2(S_2) + S_2, \quad \Omega_2 \in \mathbb{R}^{B \times N \times \frac{C}{2}} \end{aligned} \tag{5}$$

Ultimately, $\Omega_1$ and $\Omega_2$ are concatenated along the channel dimension to get the output.

**Exponential-Space Relative Position Bias.** Considering that conventional Relative Position Bias (RPB) assigns positional weights independently and overlooks the principle that nearby pixels contribute more to image generation (Liang et al., 2021), we propose the Exponential-Space Relative Position Bias (ES-RPB). Particularly, we introduce an exponential mapping to the original absolute position coordinate $(\Delta X, \Delta Y)$, which injects distance-awareness into the original RPB, thus promoting local focus. What's more, to better stabilize training, a lightweight Multi-Layer Perceptron (MLP) is employed to process the transformed coordinate $(\Delta \hat{X}, \Delta \hat{Y})$ and acquire the final bias matrix $\Lambda_E$. The operation process can be formulated as:

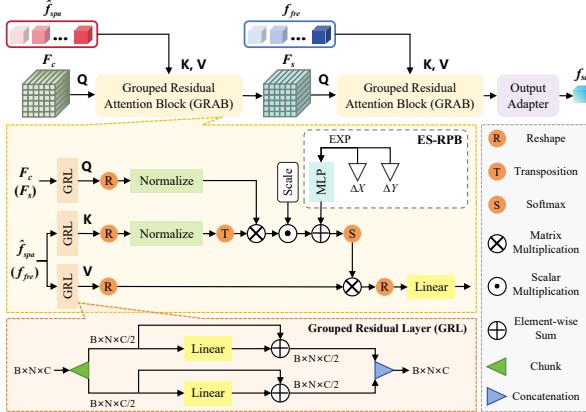

Figure 4: **Style-Content Fusion Module.** It is made up of two Grouped Residual Attention Blocks (GRABs) and one output adapter for dimensional adjustment.

$$\Delta \hat{X} = \text{sign}(\Delta X) \cdot (1 - \exp(-|\delta \cdot \Delta X|))$$
$$\Delta \hat{Y} = \text{sign}(\Delta Y) \cdot (1 - \exp(-|\eta \cdot \Delta Y|)) \quad (6)$$
$$\Lambda_E = \text{MLP}(\Delta \hat{X}, \Delta \hat{Y})$$

where $\delta$ and $\eta$ are the trainable factors that control the sensitivity to positional offsets within the same window. Besides, the symbol $\cdot$ denotes element-wise multiplication, and the MLP comprises two linear layers with an activation function in between. We set both $\delta$ and $\eta$ to 3.2 in our experiments.

Based on the GRL and ES-RPB, we elaborate on the fusion process of SCFM. Notably, to ensure dimension matching, we first flatten the spatial structure of key and value features across both blocks. In each GRAB, the query $Q$, key $K$, and value $V$ are derived by applying three GRLs to the corresponding inputs. Next, the similarity between $Q$ and $K$ is computed via normalized dot-product, which is then scaled by a learnable factor $\mu$ (initialized to 10), and further refined with $\Lambda_E$. After that, the attention weights are obtained through the Softmax function, followed by feature aggregation with $V$. Lastly, a linear projection $\Gamma_{Proj}$ is employed to output the result. The overall pipeline is represented as follows:

$$\text{GRAB} = \Gamma_{proj}(\text{Softmax}(\mu \cdot \text{Normalize}(Q) * \text{Normalize}(K)^\top + \Lambda_E)V) \quad (7)$$

In the first GRAB, the component-wise codebook $F_c$ serves as $Q$, while the refined spatial style feature $\hat{f}_{spa}$ is used as $K$ and $V$, yielding the intermediate output $F_s$. The second GRAB follows the same structure, where $Q$ is replaced by $F_s$, with both $K$ and $V$ jointly assigned to $f_{fre}$. Finally, we utilize an output adapter to adjust the channel dimension, obtaining the fused style representation $f_{sa}$. More theoretical analyses of SCFM can be found in Appendix A.5.

### 3.5 TRAINING OBJECTIVE

The loss functions of our proposed HF-Font consist of three parts: adversarial loss, matching loss, and style contrastive loss.

**Adversarial Loss.** To ensure plausibility in both style and content, we use a multi-head discriminator $D_{s,c}$ conditioned on the style label $s$ and content label $c$. The loss function is implemented on the hinge GAN loss (Zhang et al., 2019) by:

$$\mathcal{L}_{adv}^D = -\mathbb{E}_{I_t \sim p_{data}} \min(0, -1 + D_{s,c}(I_t)) - \mathbb{E}_{I_o \sim p_G} \min(0, -1 - D_{s,c}(I_o))$$
$$\mathcal{L}_{adv}^G = -\mathbb{E}_{I_o \sim p_G} D_{s,c}(I_o) \quad (8)$$

where $p_{data}$ and $p_G$ denote the set of real images and generated images, respectively.

**Matching Loss.** To mitigate mode collapse and enforce the generated character $I_o$ closely resemble to the ground truth $I_t$ at both pixel and feature levels, we apply an $\mathcal{L}_1$ loss on the image and its

features, as defined below:

$$\mathcal{L}_{img} = \mathbb{E}_{I_t \sim p_{data}} \left[ \|I_t - I_o\|_1 \right]$$

$$\mathcal{L}_{feat} = \mathbb{E}_{I_t \sim p_{data}} \left[ \sum_{m=1}^{M} \left\| D_{s,c}^{(m)}(I_t) - D_{s,c}^{(m)}(I_o) \right\|_1 \right] \tag{9}$$

where $M$ is $D_{s,c}$'s layer number and $D_{s,c}^{(m)}(\cdot)$ refers to the feature map in the $m$-th layer of $D_{s,c}$.

**Style Contrastive Loss.** To promote the extraction of discriminative style features from high-frequency information, we propose the style contrastive loss $\mathcal{L}_{cst}$. It encourages high-frequency style feature $f_{fre}$ from the same font to be closer in the feature space, while pushing apart those from different fonts, as follows:

$$\mathcal{L}_{cst} = -\frac{1}{U} \sum_{i \in \Theta} \frac{1}{|P(i)|} \sum_{p \in P(i)} \log \frac{\exp\left(v_i \cdot v_p / \tau\right)}{\sum_{a \in A(i)} \exp\left(v_i \cdot v_a / \tau\right)} \tag{10}$$

In detail, $i \in \Theta = \{1, 2, ..., U\}$ is the sample index in a batch size of $U$ and $A(i) = \Theta \backslash \{i\}$ represents other indices distinct from $i$. Besides, $v_i$ denotes the normalized high-frequency feature of the $i$-th font derived from $f_{fre}^{(i)}$. Its positive set is defined as $P(i) = \{p \in A(i) \mid y_p = y_i\}$, where $y_i$ refers to the font label of the $i$-th sample, and the remaining samples $A(i) \backslash P(i)$ serve as the negative set. Apart from these, the scalar parameter $\tau$ controls the temperature of the similarity distribution, which is set to 0.07.

**Total Loss.** Finally, we optimize HF-Font by the following full objective function:

$$\min_G \max_D (\mathcal{L}_{adv}^D + \mathcal{L}_{adv}^G + \lambda_1 \mathcal{L}_{img} + \lambda_2 \mathcal{L}_{feat} + \lambda_3 \mathcal{L}_{cst}) \tag{11}$$

Here, $\lambda_1, \lambda_2$, and $\lambda_3$ are the three weighting hyper-parameters. In our experiments, we empirically set them to 1, 1, and 0.2, respectively.

## 4 EXPERIMENTS

### 4.1 DATASET AND EVALUATION METRICS

**Dataset.** We collect a large Chinese font dataset with 575 fonts (style), each containing 3500 commonly-used Chinese characters (content) at a resolution of 128×128. Notably, the font $kai$ is fixed as the content font throughout training and testing. It is also used to pre-train the feature decomposition network for codebook acquisition.

For the training set, we randomly select 550 fonts with 3000 characters per font, forming the Seen Fonts Seen Characters (SFSC) set. Our test set includes two parts: 24 Unseen Fonts with 500 Unseen Characters per font (UFUC) and 550 Seen Fonts with 500 Unseen Characters per font (SFUC).

**Evaluation Metrics.** To comprehensively assess the quality of font generation, we utilize the following five metrics, including $L_1$ loss, Structural Similarity Index Measure (SSIM) (Wang et al., 2004), Root Mean Square Error (RMSE) (Karunasingha, 2022), Learned Perceptual Image Patch Similarity (LPIPS) (Zhang et al., 2018a), and Frechet Inception Distance (FID) (Martin et al., 2017).

Furthermore, we also perform a user study to evaluate the subjective quality of generated results. In particular, we randomly select 10 font styles from each test set and 10 characters per font. Then, we invite 20 well-educated volunteers and ask them to choose the best generation result among the comparison methods. Here, all the samples are shuffled before evaluation to avoid any potential bias. In addition, all the participants evaluate the same set of samples to ensure consistency and comparability across the results.

### 4.2 IMPLEMENTATION DETAILS

The entire training process consists of two stages. In the first stage, we train the feature decomposing network with 3000 Chinese characters rendered in the font $kai$. During this stage, we set the embedding dimension to 256, the codebook size to 100, the batch size to 64, and the number of training

Table 1: **Quantitative comparison results on UFUC and SFUC datasets.**

| Dataset | Method | Venue | SSIM↑ | RMSE↓ | LPIPS↓ | FID↓ | $L_1$↓ | User study↑ |
|---------|--------|-------|-------|-------|--------|------|--------|-------------|
| UFUC | FUNIT | ICCV 2019 | 0.6276 | 0.3356 | 0.2692 | 71.3708 | 0.1376 | 2.55% |
| | MX-Font | ICCV 2021 | 0.6966 | 0.3159 | 0.2359 | 62.7500 | 0.1235 | 5.25% |
| | DG-Font | CVPR 2021 | 0.6527 | 0.3238 | 0.2058 | 61.6022 | 0.1241 | 5.55% |
| | LF-Font | AAAI 2021 | 0.6768 | 0.3110 | 0.2516 | 66.8840 | 0.1190 | 5.45% |
| | CF-Font | CVPR 2023 | 0.6613 | 0.3102 | 0.2014 | 59.7644 | 0.1169 | 8.90% |
| | VQ-Font | ICCV 2023 | 0.6776 | 0.3066 | 0.2157 | 58.2632 | 0.1175 | 10.35% |
| | FontDiffuser | AAAI 2024 | 0.6390 | 0.3579 | 0.2816 | 79.4458 | 0.1530 | 8.35% |
| | IF-Font | NeurIPS 2024 | 0.6891 | 0.3130 | 0.2173 | 59.6292 | 0.1038 | 11.10% |
| | MX-Font++ | ICASSP 2025 | 0.7092 | 0.2897 | 0.1855 | 57.7845 | 0.0976 | 12.05% |
| | DA-Font | ACM MM 2025 | 0.7352 | 0.2854 | 0.1699 | 54.1856 | 0.0872 | 13.75% |
| | HF-Font (Ours) | - | **0.7518** | **0.2692** | **0.1544** | **50.0632** | **0.0817** | **16.70%** |
| SFUC | FUNIT | ICCV 2019 | 0.6056 | 0.3583 | 0.2777 | 68.8726 | 0.1545 | 2.25% |
| | MX-Font | ICCV 2021 | 0.6733 | 0.3314 | 0.2197 | 61.0583 | 0.1321 | 5.65% |
| | DG-Font | CVPR 2021 | 0.6478 | 0.3278 | 0.2006 | 58.3525 | 0.1277 | 5.80% |
| | LF-Font | AAAI 2021 | 0.6140 | 0.3335 | 0.2726 | 69.2239 | 0.1520 | 5.55% |
| | CF-Font | CVPR 2023 | 0.6569 | 0.3186 | 0.1974 | 57.9868 | 0.1202 | 8.60% |
| | VQ-Font | ICCV 2023 | 0.6414 | 0.3304 | 0.2069 | 56.7299 | 0.1305 | 9.95% |
| | FontDiffuser | AAAI 2024 | 0.6317 | 0.3691 | 0.2910 | 73.3052 | 0.1574 | 8.65% |
| | IF-Font | NeurIPS 2024 | 0.6652 | 0.3280 | 0.2221 | 62.8246 | 0.1213 | 10.70% |
| | MX-Font++ | ICASSP 2025 | 0.6843 | 0.3122 | 0.2043 | 57.1191 | 0.1179 | 11.40% |
| | DA-Font | ACM MM 2025 | 0.7287 | 0.3019 | 0.1792 | 49.2237 | 0.1116 | 14.05% |
| | HF-Font (Ours) | - | **0.7335** | **0.2941** | **0.1677** | **46.1692** | **0.1063** | **17.40%** |

Figure 5: **Qualitative comparison results on UFUC and SFUC datasets.** Characters in the blue boxes suffer from stroke errors (including missing or redundant strokes), while the red boxes highlight conspicuous blurring or artifacts. Zoom in to see details.

iterations to 50000. In the second stage, we train the entire model via Adam optimizer (Kingma & Ba, 2015) with $\beta_1 = 0.9$ and $\beta_2 = 0.99$. Here, the batch size is set to 8, with learning rates of $2 \times 10^{-4}$ for the generator and $4 \times 10^{-4}$ for the discriminator. The total number of iteration steps is set to 600000. For the few-shot font generation task, the number of reference images is set to 4.

## 4.3 COMPARISON WITH SOTA METHODS

We compare our proposed HF-Font with eight state-of-the-art methods, including one image-to-image translation method (FUNIT (Liu et al., 2019)) and nine classical font generation methods (MX-Font (Park et al., 2021a), DG-Font (Xie et al., 2021), LF-Font (Park et al., 2021b), CF-Font (Wang et al., 2023), VQ-Font (Pan et al., 2023), FontDiffuser (Yang et al., 2024), IF-Font (Chen et al., 2024), MX-Font++ (Wang et al., 2025), and DA-Font (Chen et al., 2025)). For a fair comparison, we retrain all these models using their default settings on our training set. To be noticed, none of these above methods explicitly leverage high-frequency information.

The quantitative comparison results are summarized in Table 1. It is evident that our proposed model achieves the best performance across all the evaluation metrics. Figure 5 displays the qualitative comparison results. We can observe that FUNIT exhibits structural incompleteness in most

Table 2: **Quantitative results on different modules.** Among them, G, H, F, and L denote the gate mechanism, high-frequency branch, SCFM, and style contrastive loss, respectively.

| Modules | | | | SSIM↑ | RMSE↓ | LPIPS↓ | FID↓ |
|---|---|---|---|---|---|---|---|
| G | H | F | L | | | | |
| | | | | 0.6434 | 0.3228 | 0.2481 | 68.7447 |
| ✓ | | | | 0.6560 | 0.3175 | 0.2293 | 63.9206 |
| ✓ | ✓ | | | 0.6953 | 0.2986 | 0.1985 | 59.1434 |
| ✓ | ✓ | ✓ | | 0.7367 | 0.2734 | 0.1679 | 54.2591 |
| ✓ | ✓ | ✓ | ✓ | **0.7518** | **0.2692** | **0.1544** | **50.0632** |

Table 3: **Quantitative results on different codebook sizes.** It demonstrates how varying the number of embeddings $F_c$ affects the quality of generated images.

| Codebook Size | SSIM↑ | RMSE↓ | LPIPS↓ | FID↓ |
|---|---|---|---|---|
| 50 | 0.6276 | 0.3675 | 0.2531 | 69.6119 |
| 75 | 0.7173 | 0.3257 | 0.2074 | 62.0768 |
| 100 | **0.7518** | **0.2692** | **0.1544** | **50.0632** |
| 125 | 0.7624 | 0.2704 | 0.1456 | 49.1751 |
| 150 | 0.7689 | 0.2698 | 0.1497 | 47.0893 |

| Content | 借鉴绩戒 | 骏咖军救 | 尽节江进 | 炕肯恐库 |
|---|---|---|---|---|
| Base Model | 借鉴绩戒 | 骏咖军救 | 尽节江进 | 炕肯恐库 |
| +G | 借鉴绩戒 | 骏咖军救 | 尽节江进 | 炕肯恐库 |
| +GH | 借鉴绩戒 | 骏咖军救 | 尽节江进 | 炕肯恐库 |
| +GHF | 借鉴绩戒 | 骏咖军救 | 尽节江进 | 炕肯恐库 |
| +GHFL | 借鉴绩戒 | 骏咖军救 | 尽节江进 | 炕肯恐库 |
| Ground Truth | 借鉴绩戒 | 骏咖军救 | 尽节江进 | 炕肯恐库 |

Figure 6: **Qualitative results on different modules.** G, H, F, and L share the same notations with Table 2. The first row is the base model. The green boxes highlight details better generated by the full model. Zoom in to see details.

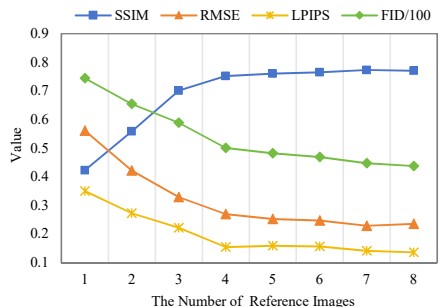

Figure 7: **Performance trend with different numbers of reference images.**

cases. Only when the source font closely resembles the target one can it generate well-layout characters. Although MX-Font and LF-Font could preserve the overall character shapes, their outputs often appear blurry with indistinct textures, resulting in degraded stroke smoothness and clarity. Besides, DG-Font also struggles to capture fine-grained details in complex glyphs, which leads to incomplete or fuzzy components. While CF-Font generally maintains the correct glyph layouts, it usually introduces noticeable artifacts. VQ-Font demonstrates good stability but lacks precise control over local features, resulting in glyph distortions and vague strokes. In addition, the performances of FontDiffuser and IF-Font are also outstanding, but their generated results sometimes suffer from obvious stroke errors. MX-Font++ improves style aggregation yet still shows obvious drawbacks in some cases, while DA-Font could preserve well structures but sometimes suffers from local blurring. As a whole, characters generated by our HF-Font are of high quality in terms of style consistency, accurate global structures, and delicate local details. Notably, Additional comparison results are provided in Appendix A.3, more generation results are included in Appendix A.6, and the cross-lingual generation results are presented in Appendix A.7.

## 4.4 ABLATION STUDY

In this section, we conduct a series of ablation experiments to access the influence of each part in our model. These experiments are performed on the UFUC dataset. More analyses and results are provided in the Appendix A.4.

**The Effect of Different Modules.** We separate the proposed modules and sequentially add them to the base model to observe their individual effects. The quantitative results are presented in Table 2, validating that all these modules can help improve the quality. Beyond numerical advances, they also bring a noticeable improvement in the visual aspects of geometric structures and stylistic strokes, as depicted in Figure 6. The gate mechanism suppresses background noise to enable more reliable spatial style extraction. Building on this, the high-frequency branch enriches local details and SCFM facilitates comprehensive feature fusion, which strengthens both structural coherence and style consistency. Moreover, the style contrastive loss further enhances the precision of results.

**The Effect of Codebook Size.** The size of codebook $F_c$ significantly affects the model's complexity and feature decomposition ability. Table 3 presents the quantitative results under different codebook sizes. We can find that the performance improves with increasing codebook size up to 100. Never-

theless, beyond this point, further enlarging the codebook yields marginal gains, with some metrics (*e.g.*, RMSE) even showing slight degradation. This suggests that while a larger codebook could enhance representation, excessive sizes would introduce more redundant information, which might compromise the model's efficiency. Hence, in other experiments, we set the codebook size to 100.

**The Effect of Reference Image Numbers.** Intuitively, given more reference images, the generated results would be better due to the availability of richer style information. As illustrated in Figure 7, the model's performance consistently improves as the number of reference images increases. Specifically, when the number rises from 1 to 4, the quality of generated results yields notable gains. However, further increases bring only marginal improvements while introducing additional inference time and memory overhead. Thus, in our paper, to strike a balance between performance and efficiency, we set the number of reference images to 4.

## 5 CONCLUSION

In this paper, we propose HF-Font, a novel few-shot font generation framework that leverages high-frequency information from reference images to enhance style extraction. To capture more distinctive style features, we introduce a style contrastive loss to facilitate the transfer process. Meanwhile, a gate mechanism is employed to suppress background noise while retaining meaningful style cues. Furthermore, we also design a Style-Content Fusion Module (SCFM) to promote effective integration of style and content representations. Both quantitative and qualitative experimental results verify that our proposed model exceeds other competitive methods on various fonts and characters.

## REPRODUCIBILITY STATEMENT

We have included detailed descriptions of our method in the main text (Section 3), as well as experimental settings (Section 4.1 and Section 4.2) to ensure reproducibility. Additional information is provided in the appendix, including the network architecture (Appendix A.1) and theoretical analyses (Appendix A.5), which further support the reproducibility of our results.

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

# A APPENDIX

In this appendix, we provide additional details and analyses, including network architecture, visualizations of the codebook, extended comparison and ablation studies, theoretical analyses of GRL and ES-RPB in SCFM, more generation results, cross-lingual generation results, as well as a brief description of our usage of large language models (LLMs) for writing polish.

## A.1 NETWORK ARCHITECTURE

Our entire model consists of two primary parts: the generator and the discriminator. Both of them are built up with two types of layers: the convolution layer and the residual layer, which are illustrated in Figure 8. Each residual layer includes two identical convolution layers. Notably, since the residual layer has its own downsampling operator, its convolution layers do not perform additional downsampling. The detailed architectures of content encoder $E_c$, spatial style encoder $E_{ss}$, high-freq style encoder $E_{sh}$, decoder $D_r$, and discriminator $D_{s,c}$ are shown in Table 4.

Table 4: **Architectures of the generator modules $E_c$, $E_{ss}$, $E_{sh}$, $D_r$, and the discriminator $D_{s,c}$.** IN and SN denote instance normalization and spectral normalization, respectively. Besides, all the padding operations are zero-padding. In the discriminator's output layer, we utilize two embedding operators to embed the output feature map into two prediction vectors of both its style and content.

| Layer Type | Normalization | Activation | Padding | Kernel Size | Stride | Downsample | Feature Maps |
|---|---|---|---|---|---|---|---|
| **Content Encoder $E_c$** | | | | | | | |
| Convolution Layer | IN | ReLU | 1 | 3 | 1 | - | 32 |
| Convolution Layer | IN | ReLU | 1 | 3 | 2 | - | 64 |
| Convolution Layer | IN | ReLU | 1 | 3 | 2 | - | 128 |
| Convolution Layer | IN | ReLU | 1 | 3 | 2 | - | 256 |
| Convolution Layer | IN | ReLU | 1 | 3 | 1 | - | 256 |
| **Spatial Style Encoder $E_{ss}$** | | | | | | | |
| Convolution Layer | IN | ReLU | 1 | 3 | 1 | - | 32 |
| Convolution Layer | IN | ReLU | 1 | 3 | 1 | AvgPool | 64 |
| Convolution Layer | IN | ReLU | 1 | 3 | 1 | AvgPool | 128 |
| Residual Layer | IN | ReLU | 1 | 3 | 1 | - | 128 |
| Residual Layer | IN | ReLU | 1 | 3 | 1 | - | 128 |
| Residual Layer | IN | ReLU | 1 | 3 | 1 | AvgPool | 256 |
| Residual Layer | IN | ReLU | 1 | 3 | 1 | - | 256 |
| Output Layer | - | Sigmoid | - | - | - | - | 256 |
| **High-freq Style Encoder $E_{sh}$** | | | | | | | |
| Convolution Layer | IN | ReLU | 1 | 3 | 1 | - | 32 |
| Convolution Layer | IN | ReLU | 1 | 3 | 1 | AvgPool | 64 |
| Convolution Layer | IN | ReLU | 1 | 3 | 1 | AvgPool | 128 |
| Residual Layer | IN | ReLU | 1 | 3 | 1 | - | 128 |
| Residual Layer | IN | ReLU | 1 | 3 | 1 | - | 128 |
| Residual Layer | IN | ReLU | 1 | 3 | 1 | AvgPool | 256 |
| Residual Layer | IN | ReLU | 1 | 3 | 1 | - | 256 |
| Output Layer | - | Sigmoid | - | - | - | - | 256 |

| Layer Type | Normalization | Activation | Padding | Kernel Size | Stride | Upsample | Feature Maps |
|---|---|---|---|---|---|---|---|
| **Decoder $D_r$** | | | | | | | |
| Residual Layer | IN | ReLU | 1 | 3 | 1 | - | 256 |
| Residual Layer | IN | ReLU | 1 | 3 | 1 | - | 256 |
| Residual Layer | IN | ReLU | 1 | 3 | 1 | - | 256 |
| Convolution Layer | IN | ReLU | 1 | 3 | 1 | Nearest | 128 |
| Convolution Layer | IN | ReLU | 1 | 3 | 1 | Nearest | 64 |
| Convolution Layer | IN | ReLU | 1 | 3 | 1 | Nearest | 32 |
| Convolution Layer | IN | ReLU | 1 | 3 | 1 | - | 1 |
| Output Layer | - | Sigmoid | - | - | - | - | 1 |
| **Discriminator $D_{s,c}$** | | | | | | | |
| Convolution Layer | SN | - | 1 | 3 | 2 | - | 32 |
| Residual Layer | SN | ReLU | 1 | 3 | 1 | AvgPool | 64 |
| Residual Layer | SN | ReLU | 1 | 3 | 1 | AvgPool | 128 |
| Residual Layer | SN | ReLU | 1 | 3 | 1 | AvgPool | 256 |
| Residual Layer | SN | ReLU | 1 | 3 | 1 | - | 512 |
| Residual Layer | SN | ReLU | 1 | 3 | 1 | - | 512 |
| Output Layer | | | | Multi-task embedding | | | |

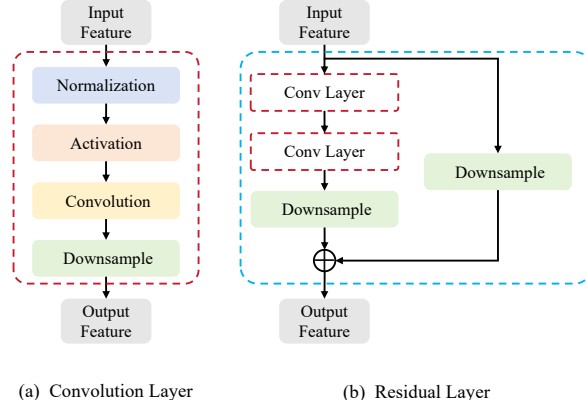

(a) Convolution Layer      (b) Residual Layer

Figure 8: **Structures of the convolution layer and the residual layer in Table 4.**

## A.2 VISUAL ANALYSES OF THE CODEBOOK

To gain a deeper understanding into the learned quantization space, we visualize some representative codebook vectors to illustrate the structural information they encode. As shown in Figure 9, each code vector exhibits distinct spatial activations, with some focusing on vertical structures and others highlighting horizontal patterns. This indicates that the codebook could implicitly capture component-level abstractions, which facilitates structural decomposition without the need for any explicit prior knowledge. Such findings provide empirical support for the interpretability of the learned codebook and its critical role in character layout modelling.

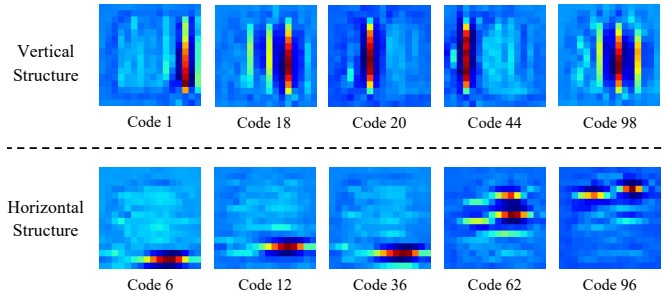

Figure 9: **The visualization of some latent codes.**

## A.3 MORE COMPARISON RESULTS

In addition to the two test sets (UFUC and SFUC) discussed in the main paper, we further conduct comparison experiments on a third benchmark, 24 Unseen Fonts with 3000 Seen Characters per font (UFSC), to evaluate the generalization ability of our proposed method.

Apart from the state-of-the-art methods mentioned in Section 4.3, we also include Diff-Font, a generative architecture built upon the diffusion model, as an extra comparison method. Notably, since Diff-Font is limited to generating seen characters, we only evaluate it on the UFSC test set.

The quantitative comparison results are presented in Table 5, which shows that our HF-Font consistently outperforms all the competing methods across multiple evaluation metrics. Figure 10 displays the qualitative comparison results. We can observe that FUNIT tends to produce structurally incomplete glyphs, while MX-Font and LF-Font preserve the overall shapes but yield blurred textures with limited visual fidelity. DG-Font struggles with local details, leading to missing strokes or misplacing components in complex characters. Besides, CF-Font maintains the overall layout but suffers from artifacts and style inconsistencies. VQ-Font achieves reasonable style adaptation yet lacks fine-grained control, resulting in glyph distortions. Although Diff-Font, FontDiffuser, and IF-Font perform relatively well, stroke-level errors still occur in some cases. MX-Font++ enhances style ag-

gregation but still exhibits noticeable issues in certain cases, whereas DA-Font generally preserves global structures yet occasionally suffers from local blurriness. In contrast, our HF-Font generates characters with superior style consistency, better structural accuracy, and refined local details.

Table 5: **Quantitative comparison results on UFSC dataset.**

| Method | Venue | SSIM↑ | RMSE↓ | LPIPS↓ | FID↓ | L1↓ | User study↑ |
|--------|-------|-------|-------|--------|------|-----|-------------|
| FUNIT | ICCV 2019 | 0.6309 | 0.3339 | 0.2685 | 59.3728 | 0.1362 | 2.40% |
| MX-Font | ICCV 2021 | 0.7062 | 0.3229 | 0.2063 | 53.3935 | 0.1205 | 4.30% |
| DG-Font | CVPR 2021 | 0.6658 | 0.3138 | 0.1877 | 50.0568 | 0.1256 | 4.70% |
| LF-Font | AAAI 2021 | 0.6812 | 0.3090 | 0.2471 | 55.2930 | 0.1176 | 4.60% |
| CF-Font | CVPR 2023 | 0.6794 | 0.3063 | 0.1836 | 49.7423 | 0.1224 | 7.95% |
| VQ-Font | ICCV 2023 | 0.6842 | 0.2995 | 0.1927 | 48.6191 | 0.1166 | 9.75% |
| Diff-Font[1] | IJCV 2024 | 0.6143 | 0.3242 | 0.1851 | 60.0310 | 0.1370 | 6.85% |
| FontDiffuser | AAAI 2024 | 0.6430 | 0.3433 | 0.2846 | 63.1833 | 0.1483 | 8.20% |
| IF-Font | NeurIPS 2024 | 0.6915 | 0.2997 | 0.1962 | 48.3104 | 0.1181 | 10.65% |
| MX-Font++ | ICASSP 2025 | 0.7351 | 0.2923 | 0.1798 | 46.7172 | 0.1127 | 11.55% |
| DA-Font | ACM MM 2025 | 0.7544 | 0.2782 | 0.1649 | 43.8763 | 0.0916 | 13.60% |
| HF-Font (Ours) | - | **0.7612** | **0.2635** | **0.1567** | **40.6694** | **0.0859** | **15.45%** |

[1] Diff-Font can only generate seen characters, so we evaluate this model on the UFSC test set alone.

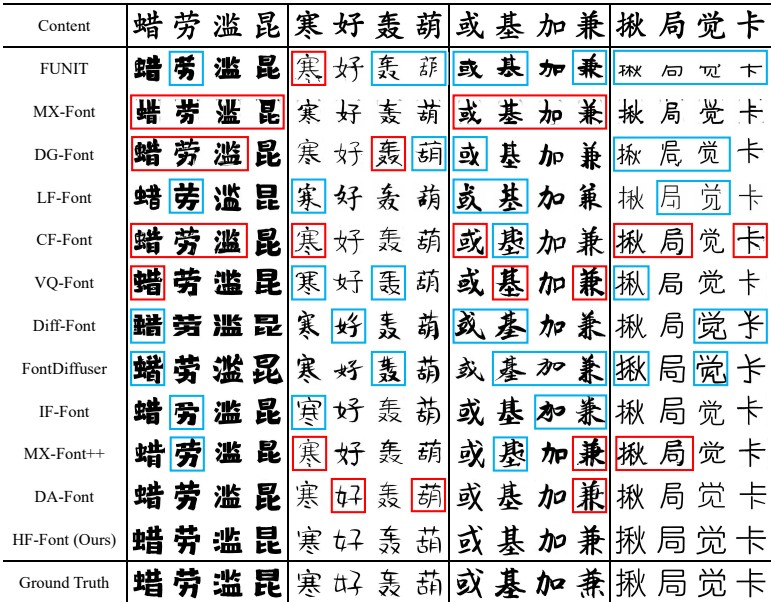

Figure 10: **Qualitative comparison results on UFSC dataset.** Characters in the blue boxes suffer from stroke errors (*e.g.*, missing or redundant strokes), while the red boxes highlight conspicuous blurring or artifacts. Zoom in to see details.

## A.4 MORE ABLATION STUDIES

Like Section 4.4, we carry out the additional ablation experiments on the UFUC dataset.

**The Effect of Different High-pass Filters.** We conduct ablation experiments to evaluate the impacts of different high-pass filters. As illustrated in Figure 11, we compare the Laplacian filter with three other commonly-used filters, including Wavelet, Fourier, and Sobel. From the figure, we notice that the Wavelet filter could capture local high-frequency details but often yields overly sparse and fragmented features, which may impede stroke continuity modelling. Besides, the Fourier filter tends to produce dim and blurry edges in the spatial domain. Additionally, although the Sobel operator provides enhancement along the gradient directions, it is susceptible to noise, resulting in disjointed contours. In contrast, the Laplacian filter extracts more accurate and sharper character contours and structures. The quantitative results in Table 6 also confirm its advantage in extracting high-frequency style features.

**The Effect of GRL and ES-RPB in SCFM.** To assess the roles of Grouped Residual Layer (GRL) and Exponential-Space Relative Position Bias (ES-RPB) in our proposed Style-Content Fusion Module (SCFM), we perform a series of ablation experiments. The quantitative results are summarized in Table 7, and the qualitative comparisons are presented in Figure 12.

From the results, we can find that removing GRL while retaining ES-RPB alone would lead to a performance drop, with SSIM decreasing from 0.7518 to 0.7327 and RMSE increasing from 0.2692 to 0.2797. This indicates that GRL facilitates effective feature fusion via structured residual learning. As shown in Figure 12, the absence of GRL results in blurred stroke details, underscoring its importance in preserving glyph topological integrity.

Similarly, disabling ES-RPB also degrades the model's performance, notably increasing LPIPS (0.1803) and FID (56.2451), which demonstrates its efficacy in spatial alignment and geometric fidelity. Without ES-RPB, the generated glyphs tend to exhibit positional distortion and weakened visual consistency.

Table 6: **Quantitative results on different high-pass filters.** It is obvious that Laplacian filter yields the best overall performance.

| Filter | SSIM↑ | RMSE↓ | LPIPS↓ | FID↓ |
|---|---|---|---|---|
| Wavelet | 0.7216 | 0.2987 | 0.1782 | 57.1143 |
| Fourier | 0.6479 | 0.3321 | 0.2236 | 66.7955 |
| Sobel | 0.6995 | 0.3103 | 0.1861 | 62.9451 |
| Laplacian | **0.7518** | **0.2692** | **0.1544** | **50.0632** |

Table 7: **Quantitative results on GRL and ES-RPB in SCFM.** The first row denotes the standard cross-attention mechanism.

| Modules | | SSIM↑ | RMSE↓ | LPIPS↓ | FID↓ |
|---|---|---|---|---|---|
| GRL | ES-RPB | | | | |
| ✗ | ✗ | 0.7152 | 0.3063 | 0.2089 | 60.0298 |
| ✓ | ✗ | 0.7284 | 0.2845 | 0.1803 | 56.2451 |
| ✗ | ✓ | 0.7327 | 0.2797 | 0.1736 | 54.4714 |
| ✓ | ✓ | **0.7518** | **0.2692** | **0.1544** | **50.0632** |

Figure 11: **Effect of different high-pass filters on style feature extraction.** Zoom in to see details.

Figure 12: **Qualitative results on GRL and ES-RPB in SCFM.** The green boxes point details that are better generated by the full model. Zoom in to see details.

A.5 THEORETICAL ANALYSES ON THE EFFECTIVENESS OF GRL AND ES-RPB IN SCFM

**Theoretical Explanation of GRL.** GRL includes two core components: the grouping scheme and the residual structure. The grouping scheme is proposed to efficiently reduce both parameters and computations without sacrificing representation performance. Meanwhile, the residual structure aids optimization by allowing the model to find the optimal solution in a residual space. The explanations are as follows:

First, we analyze the reductions of parameters and Multiply-ACcumulate operations (MACs). For a conventional linear layer with both input and output dimensions equal to $N$, the complexity in

terms of parameters and MACs amounts to $N^2$. However, by splitting the input into two parts of size $\frac{N}{2}$ and processing them with two linear layers, the total complexity decreases to $\frac{N^2}{2}$. Hence, the grouping scheme yields a clear halving of both the parameter count and computational cost.

Second, we analyze why the grouping scheme does not lead to severe performance degradation. A possible concern about the grouping scheme is that splitting channels may reduce the interactions between different groups of features. Here, we will show that, at least for query and key matrices in the attention mechanism, this worry is unnecessary. Specifically, let the query and key inputs be $X_q \in \mathbb{R}^{N_q \times C}$ and $X_k \in \mathbb{R}^{N_k \times C}$, where $N_q$ and $N_k$ denote the sequence lengths of query and key, respectively. $C$ is the channel dimension. Then, we divide each of them into two halves as follows:

$$X_q = [X_q^{(1)}, X_q^{(2)}] \qquad X_k = [X_k^{(1)}, X_k^{(2)}] \tag{12}$$

where $X_q^{(i)} \in \mathbb{R}^{N_q \times \frac{C}{2}}$ and $X_k^{(i)} \in \mathbb{R}^{N_k \times \frac{C}{2}}$. Then we apply grouped linear projections with parameter matrices $W_q^{(i)}, W_k^{(i)} \in \mathbb{R}^{\frac{C}{2} \times \frac{C}{2}}$ to derive the query matrix $Q$ and the key matrix $K$ as:

$$Q = [\, X_q^{(1)} W_q^{(1)}, \; X_q^{(2)} W_q^{(2)} \,] \qquad K = [\, X_k^{(1)} W_k^{(1)}, \; X_k^{(2)} W_k^{(2)} \,] \tag{13}$$

Next, we perform the matrix product of $Q$ and $K^T$ by:

$$\begin{aligned} QK^\top &= [\, X_q^{(1)} W_q^{(1)}, \; X_q^{(2)} W_q^{(2)} \,] \begin{bmatrix} X_k^{(1)} W_k^{(1)} \\ X_k^{(2)} W_k^{(2)} \end{bmatrix}^\top \\ &= X_q^{(1)} W_q^{(1)} (X_k^{(1)} W_k^{(1)})^\top + X_q^{(2)} W_q^{(2)} (X_k^{(2)} W_k^{(2)})^\top \end{aligned} \tag{14}$$

Among them, each element of $QK^\top$ is calculated as follows:

$$(QK^\top)_{ij} = (X_q^{(1)} W_q^{(1)})_i \cdot (X_k^{(1)} W_k^{(1)})_j + (X_q^{(2)} W_q^{(2)})_i \cdot (X_k^{(2)} W_k^{(2)})_j \tag{15}$$

From Equation 15, we can see that each value is the sum of contributions from both channel groups. Therefore, although each projection operates on a separate group of channels, the subsequent dot-product naturally aggregates information across all groups, thereby largely preventing severe performance degradation.

Third, we analyze the role of the residual structure in GRL. As discussed in Section 3.4, an attention block typically involves multiple matrix multiplications, which makes it difficult for the network to directly learn the optimal parameters. To be specific, given the input of attention mechanism as $X$ and the desired projections as $\widehat{Q}$, $\widehat{K}$, and $\widehat{V}$, the network without residual connections must learn these mappings entirely from scratch. In contrast, when introducing a residual structure for the QKV linear layers, the network only needs to model $\widehat{Q} - X$, $\widehat{K} - X$, and $\widehat{V} - X$. This transformation effectively shifts the optimization from the original linear space to a residual space, which tends to be smoother and easier to navigate. Consequently, the residual design not only facilitates more efficient optimization but also enhances the representational capacity of the QKV layers.

**Theoretical Explanation of ES-RPB.** To better understand the spatial prior introduced by ES-RPB, we provide a visual comparison between linear and exponential coordinate mappings used in position encoding. As illustrated in Figure 13, the exponential transformation maps the absolute positional coordinates $(\Delta X, \Delta Y)$ from linear space to exponential space $(\Delta \widehat{X}, \Delta \widehat{Y})$, thereby increasing spatial sensitivity near the centre. This design reflects a distance-aware inductive bias that closer pixels should contribute more to attention, which is a desirable property for spatial alignment in the font generation task. Apart from this, ES-RPB also incorporates two other improvements. First, it replaces direct parameter learning with a lightweight MLP to minimize the noise impact and enhance training stability. Second, instead of learning relative positional parameters in isolation, ES-RPB models them within the shared functional space of the MLP, which encourages richer interactions and better generalization across spatial contexts.

## A.6 MORE GENERATION RESULTS

To further validate the robustness and generalization capability of our proposed HF-Font, we conduct two extra groups of experiments. The first group investigates the model's performance across a

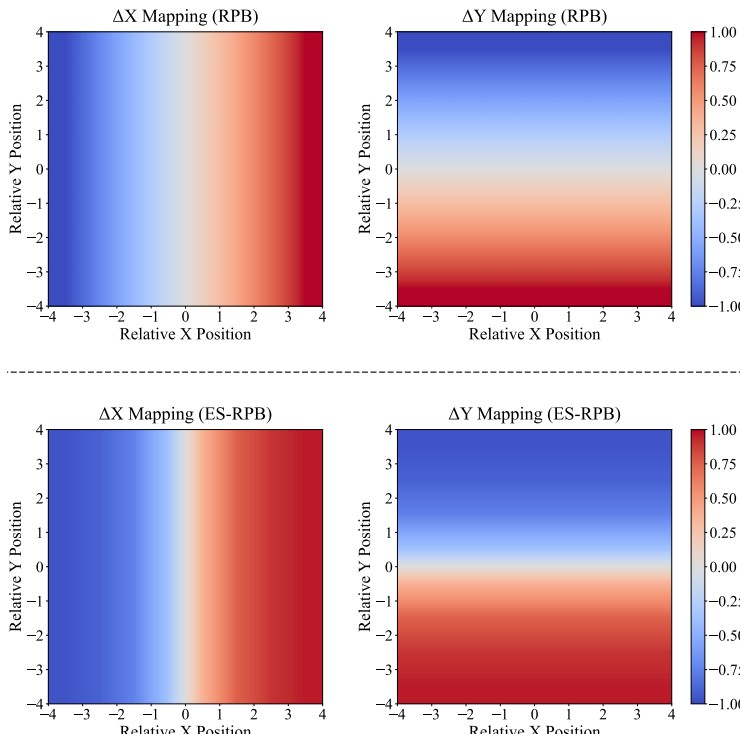

Figure 13: **Comparison between coordinate mapping functions used in RPB and ES-RPB.** The top row shows linear normalized mapping in the X-direction (left) and Y-direction (right). The bottom row displays exponential-space mapping, where changes near the centre are more prominent. The colour scale represents the relative position bias values, ranging from -1 (blue) to 1 (red).

wide range of diverse and stylistically rich typefaces, thereby aiming to assess its adaptability to substantial style variations (Figure 14). The second group targets the generation of structurally complex characters, which contain dense strokes and intricate component arrangements, to verify the model's ability in preserving fine-grained structural fidelity under challenging cases (Figure 15). Experimental results from both settings consistently demonstrate the effectiveness and reliability of our approach in handling a broad spectrum of font generation scenarios.

### A.7 CROSS-LINGUAL FONT GENERATION

An interesting question is whether our method can be applied to the generation of other Eastern Asian font styles, such as Japanese Kana and Korean Hangul. Obviously, collecting a large number of character images in these font styles for model training directly would fulfil this goal but also be highly time-consuming and labour-intensive. On the other hand, considering the structural and visual similarities among these font styles, we thus utilize the framework trained on our collected Chinese fonts to extend the generation of other font styles directly. To be specific, we randomly choose 15 fonts with 100 Japanese Kana characters and 10 fonts with 125 Korean Hangul characters for testing, respectively. Some generation results are displayed in Figure 16. We can see that since the structures of the Japanese Kana and Korean Hangul characters are simpler than those of Chinese characters, the generation results closely resemble the ground truths, demonstrating the robust cross-domain generalization ability of our proposed model.

### A.8 USAGE OF LARGE LANGUAGE MODELS

During manuscript preparation, we employed several large language models (such as ChatGPT and DeepSeek) solely for the purpose of polishing the writing and improving readability. These models were not used for research ideation, experiment design, data analysis, or result interpretation. The authors take full responsibility for the content of the paper.

| Content | 赖 览 兰 况 款 苦 | 立 类 牢 扣 克 孔 | 帘 良 荔 愕 梨 联 | 两 老 例 连 丽 吏 |
|---|---|---|---|---|
| Generated Results | 赖 览 兰 况 款 苦 | 立 类 牢 扣 克 孔 | 帘 良 荔 愕 梨 联 | 两 老 例 连 丽 吏 |
| Ground Truth | 赖 览 兰 况 款 苦 | 立 类 牢 扣 克 孔 | 帘 良 荔 愕 梨 联 | 两 老 例 连 丽 吏 |
| Content | 坑 唠 廊 李 了 里 | 几 回 昏 化 卉 吉 | 劣 将 坚 轿 秸 荐 | 练 亮 疗 伶 粒 恋 |
| Generated Results | 坑 唠 廊 李 了 里 | 几 回 昏 化 卉 吉 | 劣 将 坚 轿 秸 荐 | 练 亮 疗 伶 粒 恋 |
| Ground Truth | 坑 唠 廊 李 了 里 | 几 回 昏 化 卉 吉 | 劣 将 坚 轿 秸 荐 | 练 亮 疗 伶 粒 恋 |
| Content | 帝 测 炒 碘 带 财 | 景 姐 见 讲 杰 决 | 革 丰 肤 父 芬 盖 | 可 看 开 凯 具 刊 |
| Generated Results | 帝 测 炒 碘 带 财 | 景 姐 见 讲 杰 决 | 革 丰 肤 父 芬 盖 | 可 看 开 凯 具 刊 |
| Ground Truth | 帝 测 炒 碘 带 财 | 景 姐 见 讲 杰 决 | 革 丰 肤 父 芬 盖 | 可 看 开 凯 具 刊 |
| Content | 邻 裂 临 料 僚 猎 | 固 瓜 刮 怪 关 归 | 怯 困 宽 刻 考 灸 | 副 柑 哥 工 公 勾 |
| Generated Results | 邻 裂 临 料 僚 猎 | 固 瓜 刮 怪 关 归 | 怯 困 宽 刻 考 灸 | 副 柑 哥 工 公 勾 |
| Ground Truth | 邻 裂 临 料 僚 猎 | 固 瓜 刮 怪 关 归 | 怯 困 宽 刻 考 灸 | 副 柑 哥 工 公 勾 |
| Content | 汇 黄 宦 画 沪 忽 | 奋 伏 负 妇 羔 告 | 佳 剂 频 鸡 祸 混 | 吵 忱 持 恬 度 饭 |
| Generated Results | 汇 黄 宦 画 沪 忽 | 奋 伏 负 妇 羔 告 | 佳 剂 频 鸡 祸 混 | 吵 忱 持 恬 度 饭 |
| Ground Truth | 汇 黄 宦 画 沪 忽 | 奋 伏 负 妇 羔 告 | 佳 剂 频 鸡 祸 混 | 吵 忱 持 恬 度 饭 |

Figure 14: **More generation results on various font styles.** The first row shows the content images, the second row denotes the characters generated by our HF-Font, and the third row represents the ground truth. Zoom in to see details.

| Content | 塞 鲨 骚 蕊 赛 撒 | 燃 鹊 壤 熔 融 踩 | 蠕 褥 裙 趣 锹 儒 | 搂 擎 瘸 嚷 瓢 擒 |
|---|---|---|---|---|
| Generated Results | 塞 鲨 骚 蕊 赛 撒 | 燃 鹊 壤 熔 融 踩 | 蠕 褥 裙 趣 锹 儒 | 搂 擎 瘸 嚷 瓢 擒 |
| Ground Truth | 塞 鲨 骚 蕊 赛 撒 | 燃 鹊 壤 熔 融 踩 | 蠕 褥 裙 趣 锹 儒 | 搂 擎 瘸 嚷 瓢 擒 |
| Content | 厦 雯 赡 擅 裳 摄 | 煽 甥 盛 誓 嗜 匙 | 剩 赚 嘱 骤 撞 撰 | 臊 烧 瘦 淑 释 输 |
| Generated Results | 厦 雯 赡 擅 裳 摄 | 煽 甥 盛 誓 嗜 匙 | 剩 赚 嘱 骤 撞 撰 | 臊 烧 瘦 淑 释 输 |
| Ground Truth | 厦 雯 赡 擅 裳 摄 | 煽 甥 盛 誓 嗜 匙 | 剩 赚 嘱 骤 撞 撰 | 臊 烧 瘦 淑 释 输 |
| Content | 瘫 檀 糖 藤 蹄 瞳 | 撕 艘 髓 嗽 穗 酸 | 鼠 蹋 嗓 肆 踏 嘶 | 蔬 曙 蟀 霜 瞬 寝 |
| Generated Results | 瘫 檀 糖 藤 蹄 瞳 | 撕 艘 髓 嗽 穗 酸 | 鼠 蹋 嗓 肆 踏 嘶 | 蔬 曙 蟀 霜 瞬 寝 |
| Ground Truth | 瘫 檀 糖 藤 蹄 瞳 | 撕 艘 髓 嗽 穗 酸 | 鼠 蹋 嗓 肆 踏 嘶 | 蔬 曙 蟀 霜 瞬 寝 |
| Content | 躺 腾 艇 褪 臀 豌 | 蔚 瘟 嗡 菱 趟 慰 | 蜀 隧 潭 巍 薇 腕 | 睡 鹅 舞 题 雾 魏 |
| Generated Results | 躺 腾 艇 褪 臀 豌 | 蔚 瘟 嗡 菱 趟 慰 | 蜀 隧 潭 巍 薇 腕 | 睡 鹅 舞 题 雾 魏 |
| Ground Truth | 躺 腾 艇 褪 臀 豌 | 蔚 瘟 嗡 菱 趟 慰 | 蜀 隧 潭 巍 薇 腕 | 睡 鹅 舞 题 雾 魏 |
| Content | 箫 蟹 薪 醒 熊 墟 | 橡 潇 蝎 懈 馨 歇 | 酗 癣 锈 薛 熏 蜓 | 膝 霞 嘻 嫌 镶 袭 |
| Generated Results | 箫 蟹 薪 醒 熊 墟 | 橡 潇 蝎 懈 馨 歇 | 酗 癣 锈 薛 熏 蜓 | 膝 霞 嘻 嫌 镶 袭 |
| Ground Truth | 箫 蟹 薪 醒 熊 墟 | 橡 潇 蝎 懈 馨 歇 | 酗 癣 锈 薛 熏 蜓 | 膝 霞 嘻 嫌 镶 袭 |

Figure 15: **More generation results on complex characters.** The first row shows the content images, the second row denotes the characters generated by our HF-Font, and the third row represents the ground truth. Zoom in to see details.

| | | | | |
|---|---|---|---|---|
| Content | ぐ せ み む り ン | ね の は ほ み テ | ぎ ざ だ づ で ス | ご じ げ ぜ ぞ に |
| Generated Results | ぐ せ み む り ン | ね の は ほ み テ | ぎ ざ だ づ で ス | げ ご じ ぜ ぞ に |
| Ground Truth | ぐ せ み む り ン | ね の は ほ み テ | ぎ ざ だ づ で ス | げ ご じ ぜ ぞ に |
| Content | め や も よ ら な | た ぢ つ て ぬ を | ヤ ユ ヨ ラ リ せ | ど ば び ぶ ぼ す |
| Generated Results | め や も よ ら な | た ぢ つ て ぬ を | ヤ ユ ヨ ラ リ せ | ど ば び ぶ ぼ す |
| Ground Truth | め や も よ ら な | た ぢ つ て ぬ を | ヤ ユ ヨ ラ リ せ | ど ば び ぶ ぼ す |

**(a) Generated Results on Japanese (Kana) characters**

| | | | | |
|---|---|---|---|---|
| Content | 교 곤 콤 콥 곳 게 | 괘 괏 팹 광 괜 귀 | 갉 갋 값 감 갑 젝 | 괴 괵 괸 굄 굅 갸 |
| Generated Results | 교 곤 콤 콥 곳 게 | 괘 괏 팹 광 괜 귀 | 갉 갋 값 감 갑 젝 | 괴 괵 괸 굄 굅 갸 |
| Ground Truth | 교 곤 콤 콥 곳 게 | 괘 괏 팹 광 괜 귀 | 갉 갋 값 감 갑 젝 | 괴 괵 괸 굄 굅 갸 |
| Content | 구 국 군 굳 굼 괌 | 규 그 극 근 귿 걍 | 가 각 간 갇 갈 과 | 값 갓 갔 강 갖 권 |
| Generated Results | 구 국 군 굳 굼 괌 | 규 그 극 근 귿 걍 | 가 각 간 갇 갈 과 | 값 갓 갔 강 갖 권 |
| Ground Truth | 구 국 군 굳 굼 괌 | 규 그 극 근 귿 걍 | 가 각 간 갇 갈 과 | 값 갓 갔 강 갖 권 |

**(b) Generated Results on Korean (Hangul) characters**

Figure 16: **Generation results on cross-lingual scenario.** In this case, we generate some Japanese Kana and Korean Hangul glyph images with the model trained on Chinese fonts.