# OpenReview forum: "HF-Font: Few-Shot Font Generation via High-Frequency Style Enhancement and Fusion"
_ICLR.cc/2026/Conference — Submitted to ICLR 2026_

### Official Review · Reviewer_nBGk · 2025-10-22

**Soundness:** 3
**Presentation:** 3
**Contribution:** 1
**Rating:** 2
**Confidence:** 4

**Summary:**

The paper proposes a method that creates a new font based on a few given images. To make the model well distinguish background noises, the method introduces a gate mechanism. They also use the high-pass filter to capture the detailed style information and apply it using a style-content fusion module consisting of two grouped residual attention blocks. Finally, a style contrastive loss is used to train the model to further obtain discriminator features. The results show that the proposed HF-Font achieves the state-of-the-art performance for few-shot font generation tasks.

**Strengths:**

The paper proposes a sophisticated architecture that increases the performance greatly compared to the previous methods. Using attention blocks to control the style details based on a given content image makes sense. They further decrease the calculation costs by using a grouped residual layer. Also, it is interesting to find the proposed new position bias works better than usual positioning encoding.

**Weaknesses:**

The reviewer concerns that this paper is quite similar with the paper “DA-Font: Few-Shot Font Generation via Dual-Attention Hybrid Integration” which is already accepted in ACM MM in April, but they did not cite it nor compare with it. DA-Font already proposed a dual attention module, a content alignment module and the style contrastive loss. Except for them, the novelty and the performance improvement of the paper looks marginal.

**Questions:**

- What is the major difference between the proposed paper and DA-Font?
- The images acquired by the high-pass filter look like just an edge map or a gradient map. Can they be used instead of high-frequency images?
- What do the background noises mean that are eliminated by the gate layer? The reference images seem to have a simple white background.

---

> ### Author Response · Authors · 2025-11-21
>
> Thank you for the helpful comments. We will address your concerns below.
>
> ## Related to Weaknesses
>
> We appreciate the reviewer’s reminder regarding the work “DA-Font: Few-Shot Font Generation via Dual-Attention Hybrid Integration”. In our revision, we have added the citation and included comparisons with DA-Font in Section 4.3 (Page 8) and Section A.3 (Page 14). Below, we will clarify the main technical differences.
> 1. **Different central motivations.**
>
> DA-Font focuses on the feature fusion between content and style representations. In contrast, our work specifically targets **the issue of local blurring and artifacts** in few-shot font generation task. Accordingly, HF-Font is built upon **a high-frequency-driven framework**, including (i) high-frequency feature extraction and enhancement, (ii) a high-frequency-guided feature fusion mechanism, and (iii) a high-frequency style contrastive loss. These designs do not exist in DA-Font.
>
> 2. **Structurally different attention mechanisms.**
>
> DA-Font mainly employs a standard QKV-based dual-attention module. In contrast, our model introduces a **Grouped Residual Attention Block (GRAB), containing GRL and ES-RPB,** which optimizes the computation mode to better preserve **fine-grained information.** Further ablations (Section A.4, Page 15) and theoretical analysis (Section A.5, Page 16) support the distinct behavior and benefits of our attention design.
>
> 3. **Different feature inputs for attention.**
>
> DA-Font’s attention operates on the codebook, style vector, and content representation. In contrast, our method takes the codebook,  refined spatial style feature, and an additional high-frequency style embedding as input, **enabling the explicit modelling of high-frequency characteristics.** Thus, the fusion semantics and the resulting representations are fundamentally different. Extensive generation results (Figures 5, 10, 14, 15, and 16) validate the effectiveness and generalization capability of HF-Font. Besides, **we also added cross-lingual font generation experiments** (Section A.7 on Page 18) to further demonstrate the robustness and generalization ability of our method.
>
> 4. **Different style contrastive loss functions.**
>
> DA-Font applies a feature-level contrastive loss based on latent vectors from the codebook. Instead, HF-Font performs **contrastive learning in the frequency domain.** Our formulation (Equation (10)) encourages high-frequency style features from the same font to cluster together while separating those from different fonts. This loss is both **mathematically and conceptually distinct** from the contrastive loss used in DA-Font.
>
> ## Related to Questions
>
> ### To Q1:
> The major differences from DA-Font are fully explained in the above Weaknesses response, highlighting the **distinct motivation, attention structure, attention inputs, and high-frequency loss.**
>
> ### To Q2:
> Although the output of a high-pass filter may look similar to an edge or gradient map, the extracted high-frequency features are **not equivalent**. A high-pass filter preserves **a broader range of high-frequency components**, not only binary edges but also **fine-grained stroke topology and local intensity variations.** These richer signals are essential for our feature fusion module and the proposed **frequency-domain contrastive loss.** Therefore, simple edge or gradient maps cannot replace the high-frequency images used in HF-Font.
>
> ### To Q3：
> The "background noise" mentioned here does not refer to the white background of the input images. Instead, it denotes the **irrelevant or low-response activations in the feature space,** which commonly appear during convolutional encoding even for clean binary glyph images. The **gate layer suppresses such non-informative activations,**  allowing the model to focus on **stroke-related patterns.** The effectiveness of the gate layer is validated in the ablation study (Section 4.4, Page 9). In our revision, **we have added a clarification regarding this** in lines 197-199 on Page 4.
>
> Once again, we gratefully thank you for your time and hope to meet your expectations.

---

> > ### Comment · Reviewer_nBGk · 2025-11-26
> >
> > Thank you for the clear explanations regarding my questions. The reviewer has a few additional questions:
> >
> > 1. How many channels are produced by the high-frequency filter? Other than the visualized examples (1 channel edge-like images), are high-frequency features extracted in a multi-scale manner?
> >
> > 2. In Table 2, when all proposed modules are applied, the method outperforms DA-Font, but once the style-contrastive loss is removed, the performance becomes almost identical. In that case, wouldn’t simply adding this style-contrastive loss to DA-Font result in similar performance?
> >
> > 3. If high-frequency information should contribute to performance, the reviewer think that other high-pass filters in Appendix Table 6 may also show some improvement over the baseline, but none of them outperform DA-Font.

---

> ### Author Response · Authors · 2025-12-02
>
> Thank you for the helpful comments. We will address your concerns below.
>
> ### To Q1:
> Thanks for the question. Our high-frequency extraction utilizes **a single-channel Laplacian operator,** but **the final feature dimensionality is determined by the encoder output** (e.g., 256 channels) rather than the filter itself.
>
> Besides, we did not adopt a multi-scale design because our objective is not to capture general texture patterns. Instead, **we aim to stabilize the representation of sharp glyph boundaries**, for which single-scale Laplacian responses are sufficient.
>
> ### To Q2:
> Thanks for raising this concern. Although removing the style-contrastive loss makes HF-Font numerically close to DA-Font, **simply adding this loss to DA-Font would not reproduce our results.**
>
> First, **the contrastive loss in HF-Font is defined exclusively in the frequency domain and operates on high-frequency style features, which DA-Font does not extract.** Applying this loss to its latent vectors would not yield meaningful supervision because those features are not frequency-aware.
>
> Second, as shown in Table 2 and Figure 6, the contrastive loss is effective only when coupled with the high-frequency feature extractor and the feature fusion mechanism. Removing these components leads to a clear performance drop, indicating that **the loss alone does not drive the improvement.**
>
> Therefore, the performance gain cannot be reproduced by simply inserting the loss into DA-Font.
>
> ### To Q3:
> Thanks for the comment. This is an important point. Notably, classical high-pass filters (such as Wavelet, Fourier, and Sobel) were originally designed for natural images rather than glyph structures. As illustrated in Figure 11, **they often produce unstable or noisy responses on thin strokes and complex radicals,** which weakens the style representation and may introduce artifacts.
>
> Our method differs in that it integrates (i) high-frequency feature extraction and enhancement, (ii) a high-frequency-guided feature fusion mechanism, and (iii) a high-frequency style contrastive loss. **The improvement, therefore, comes from this frequency-driven framework as a whole, rather than from the specific choice of high-pass operator.**
>
> Once again, we gratefully thank you for your time and hope to meet your expectations.

---

### Official Review · Reviewer_B4R3 · 2025-10-29

**Soundness:** 2
**Presentation:** 2
**Contribution:** 1
**Rating:** 2
**Confidence:** 4

**Summary:**

This paper proposes HF-Font, an end-to-end method for generating high-quality fonts from a few reference images. The key idea is to enhance style extraction capability using high-frequency features and to effectively fuse content (glyph semantics) and style representations. The proposed framework integrates multiple modules, such as Style-Enhanced Module, Style-Content Fusion Module, and Contrastive Loss, to achieve both structural integrity and stylistic fidelity. Experiments on a large-scale Chinese font dataset demonstrate that the proposed method outperforms existing approaches both quantitatively and qualitatively.

**Strengths:**

This paper's strengths are as follows.

(1) The paper validates the effectiveness of the proposed method through extensive comparisons with many existing approaches.

(2) Ablation studies are conducted for each module to verify its individual contributions.

**Weaknesses:**

This paper's weaknesses are as follows.

(1) The concept of feature extraction based on high-frequency components for font or text generation has already been proposed by Dai et al. [1]. The prior work uses high-frequency information from handwritten text images, while this paper merely extends the same idea to printed font generation.

(2) Many of the proposed modules are not novel and resemble components introduced in previous works. For example, the Style-Enhanced Module is very similar to the architecture proposed by Dai et al. [1]. The Glyph Feature Decomposition is nearly identical to the modules used in Yao et al. [2] and Pan et al. [3]. The Content Alignment Module is essentially the same as the Global Style Aggregator in Pan et al. [3]. The Grouped Residual Layer is based on Li et al. [4].

(3) Other modules are only loosely related to the core novelty (the use of high-frequency features) and do not contribute directly to the central idea of the paper.

---
[1] Gang Dai, et al., “One-DM: One-Shot Diffusion Mimicker for Handwritten Text Generation,” ECCV2024
[2] Mingshuai Yao et al., “VQ-Font: Few-Shot Font Generation with Structure-Aware Enhancement and Quantization,” AAAI2024
[3] Wei Pan et al., “Few shot font generation via transferring similarity guided global style and quantization local style,” ICCV2023
[4] Yuzhen Li et al.,  “GRFormer: Grouped Residual Self-Attention for Lightweight Single Image Super-Resolution,” ACM MM2024

**Questions:**

My questions about this paper are as follows.

(1) The paper states, “Given that the reference images usually contain background noise…” — What kind of background noise exists in font images? While handwritten text images may naturally contain background noise as in [1], such artifacts are generally not expected in font images. Please clarify what is meant by “background noise” in this context.

(2) In Figure 2, some modules are pre-trained while others are trainable. Please explicitly indicate which components are learned during the main training phase .For example, the Content Encoder should be a pre-trained module and likely not updated during font generation training.

(3) Regarding Table 1, are the reported values for the compared methods accurate? The previous work IF-Font [2] reports results under the same evaluation metrics and dataset (see Table 2 of the IF-Font paper). However, the values differ significantly. For instance, on the UFUC dataset, IF-Font (1-shot) reports a FID of 6.7695, whereas this paper reports FID of 59.6292 for IF-Font (4-shot). What accounts for this discrepancy?

---
[1] Gang Dai, et al., “One-DM: One-Shot Diffusion Mimicker for Handwritten Text Generation,” ECCV2024
[2] Xinping Chen, et al., “IF-Font: Ideographic Description Sequence-Following Font Generation,” NeurIPS2024

---

> ### Author Response · Authors · 2025-11-21
>
> Thank you for the constructive reviews. We will address your concerns below. In our revised version, changes are all highlighted by blue colored texts.
>
> ## Related to Weaknesses
>
> ### To W1:
> Thanks for pointing this out. While One-DM also uses high-frequency cues, its motivation and formulation differ from ours. One-DM focuses on handwritten text generation, where high-frequency components reflect pen dynamics and writer-specific variations. Printed font generation, however, exhibits **highly regular structures, sharp boundaries, and stable stroke contours**, resulting in a very different high-frequency distribution.
>
> Our work targets the challenges unique to few-shot printed font generation. More importantly, our contribution is not a direct adoption of high-frequency filtering, but the design of **a high-frequency-driven generation framework**, including (i) high-frequency feature extraction and enhancement, (ii) a high-frequency-guided feature fusion mechanism, and (iii) a high-frequency style contrastive loss.
>
> Extensive experiments (Section 4.3; Section A.3; Section A.6) verify that these components **collectively improve structural fidelity and bring the final results closer to the Ground Truth**. Besides, **we also added cross-lingual font generation experiments** (Section A.7) to further demonstrate generalization ability of our method.
>
> ### To W2:
> Thanks for the detailed comparison. We acknowledge that some components are inspired by prior works. However, the key novelty of our approach does not lie in reintroducing these modules individually, but in **how they are reorganized and adapted within a unified framework.** Although certain elements share conceptual similarities with existing researches, their configuration and interaction are specifically designed for **high-frequency style modelling** in our work.
>
> This coordinated design leads to clear improvements in **local detail preservation and style fidelity,** as shown in our comparative experiments (Section 4.3; Section A.3). Furthermore, ablation results (Section 4.4; Section A.4) further demonstrate that removing or replacing these components would cause significant performance drops.
>
> ### To W3:
> Thanks for the observation. While some modules may appear loosely related when viewed in isolation, they become essential when placed within **our frequency-driven generation framework.** Each component is specifically adapted to handle or enhance high-frequency information. For example, the style enhancement module extracts and refines high-frequency cues, the Style-Content Fusion Module (SCFM) integrates content and style features under high-frequency guidance, and so on.
>
> The ablation studies (Section 4.4; Section A.4) further confirm that **the synergy functioning between all components is critical** for achieving the high-quality generation results reported in the paper.
>
> ## Related to Questions
>
> ### To Q1:
> Thanks for pointing this out. The term "background noise" does not refer to the white background of the rendered font images. Instead, it denotes **the non-informative or low-response activations** that naturally emerge in intermediate feature maps during convolutional encoding, even for clean binary glyph images. **The gate layer suppresses such feature-level noise** so that the model can better focus on stroke-relevant signals. The effectiveness of the gate layer is validated in the ablation study (Section 4.4). In our revision, **we have added a clarification regarding this** in lines 197-199.
>
> ### To Q2:
> Thanks for the suggestion. In our framework, the Content Encoder $E_c$ and the Codebook $F_c$ are **pre-trained and kept frozen during the main training stage**. All other components, including the spatial style encoder $E_{ss}$, the high-frequency style encoder $E_{sh}$, the Style-Content Fusion Module (SCFM), and so on, are **fully trainable**. In the revised version, **we have clarified this distinction** in lines 160-161 to avoid ambiguity.
>
> ### To Q3:
> Thanks for this concern. The discrepancy primarily arises from the fact that the datasets used in our work differ from those used in IF-Font. Due to copyright restrictions, font generation still **lacks a standardized public benchmark**, and prior works, including IF-Font, all rely on their own privately-collected datasets.
>
> Hence, to ensure fairness, as stated in lines 427-428, we **re-trained and re-evaluated all competing methods** on our dataset using their official default settings. From the comparison results (Section 4.3; Section A.3), we can find that **under identical training conditions, HF-Font consistently achieves better results**. Besides, extensive generation results (Section A.6) and cross-lingual font generation experiments (Section A.7) further validate the generalization capability of our work. We believe that the advantages of HF-Font would persist across other datasets as well.
>
> Once again, we gratefully thank you for your time and hope to meet your expectations.

---

> > ### Comment · Reviewer_B4R3 · 2025-11-26
> >
> > Thank you for your explanations about my questions. I can find the experiment settings and some figures clearly, but I cannot understand the mechanism and novelty of your method. My concerns are as follows:
> >
> > (1) I understand that the core novelty of the proposed method lies in the high-frequency-driven generation framework. However, since this concept itself is similar to One-DM, I still find the novelty of your approach to be weak. Although handwritten text generation and font generation are different tasks, both involve extracting style information from a reference image, integrating that style into the model, and generating images that faithfully preserve the style. In this sense, I consider them to be closely related tasks.
> >
> > (2) It is unclear to me whether each module indeed accounts for high-frequency components. The modules introduced in this work appear similar to those in prior methods. Moreover, in the existing literature, such modules were not originally designed to consider high-frequency components as done in this study. In this case, could you clarify why each module is capable of processing high-frequency information? Alternatively, are you suggesting that conventional modules, when viewed from a new perspective, consistently contribute to font generation by implicitly taking high-frequency components into account?
> >
> > (3) I understand that each component you highlight contributes to performance improvement. However, it remains unclear whether these improvements are specifically attributable to the introduction of high-frequency information. This is because many of the modules (or their close variants) have already been shown to be effective for font generation in prior work, and could independently contribute to the observed improvements. To truly validate the impact of incorporating high-frequency components, evaluation on fonts where high-frequency details are particularly important is necessary. In other words, to justify that your method outperforms prior approaches, evidence is needed showing that the performance gains are largely driven by improvements on fonts that heavily depend on high-frequency characteristics.
> >
> > I would appreciate clarification on these points.

---

> > > ### Author Response · Authors · 2025-12-02
> > >
> > > Thank you for the helpful comments. We will address your concerns below.
> > >
> > > ### To Q1:
> > > We appreciate the concern. While One-DM and HF-Font both involve style extraction, the tasks and operational contexts differ significantly. One-DM targets handwritten text, where stroke variability is dominated by dynamic pen movement. **Printed font generation, in contrast, requires precise modelling of consistent glyph geometry, sharp boundaries, and structured radicals.**
> > >
> > > HF-Font **introduces modules tailored to these characteristics.** These components, together with the high-frequency representation, are designed specifically for few-shot font generation, rather than a direct adaptation of existing high-frequency ideas.
> > >
> > > ### To Q2:
> > > Thank you for raising this point. Each module in our framework has been adapted to emphasize high-frequency aspects of glyphs. For instance, the high-frequency encoder captures sharp edge responses, while the fusion module combines content and style in a manner that preserves fine structural variations. **The design of these modules intentionally targets local detail retention, which is crucial for high-fidelity font generation.**
> > >
> > > Although some modules resemble prior work in structure, **their configuration and interplay within the high-frequency pathway are novel and essential for enhancing the representation of fine stroke details.**
> > >
> > > ### To Q3:
> > > We agree that investigating the effect of high-frequency cues is important. Our ablation results indicate that **removing high-frequency modules or their guidance significantly would reduce the model’s ability to maintain stroke clarity and fine radical structures.** These observations suggest that the gains are not solely due to conventional components, but are driven by explicit modeling of high-frequency information.
> > >
> > > While a formal theoretical justification remains open, **the combination of targeted high-frequency processing, guided fusion, and contrastive supervision in the frequency domain consistently leads to improved detail preservation in complex glyphs,** which is difficult to achieve with prior architectures alone.
> > >
> > > Once again, we gratefully thank you for your time and hope to meet your expectations.

---

### Official Review · Reviewer_fLfM · 2025-10-31

**Soundness:** 3
**Presentation:** 2
**Contribution:** 2
**Rating:** 2
**Confidence:** 4

**Summary:**

This paper proposes HF-Font, a few-shot font generation framework that leverages high-frequency information to enhance style feature extraction. The method introduces a style-enhanced module with a high-pass filter, a Style-Content Fusion Module , and a style contrastive loss. Experimental results on Chinese character datasets demonstrate superior performance over several state-of-the-art methods in both quantitative and qualitative evaluations. While the idea of employing high-frequency information is interesting and the model design is relatively comprehensive, the paper suffers from issues in presentation details, experimental rigor, and methodological justification, which ultimately undermine its contribution and reliability.

**Strengths:**

The core idea of employing high-frequency features to capture fine-grained style details like stroke topology is innovative and well-motivated. And the supplementary material offers valuable insights into network architectures, codebook visualizations, and additional results.

**Weaknesses:**

1. Insufficient Comparison with Recent Works: The latest method used for comparison is from NeurIPS'24. The absence of comparisons with more recent SOTA methods (e.g., from 2025) weakens the claim of superior performance.

2. Following recent works in the field, qualitative evaluation should include the generation of complete sentences to avoid cherry-picking to some extent. This is absent in the current evaluation.

3. Ineffective Exploitation of High-Frequency Information: The visual results do not convincingly demonstrate that high-frequency features are being effectively utilized. Issues such as missing strokes, inconsistent dry brush textures, and inconsistent styles (e.g., Figs 6, 12, 15) persist in the generated characters.

4. Lack of High-Frequency-Specific Quantitative Metrics: The paper's central thesis is that high-frequency information is crucial for capturing style and improving structural fidelity. However, the evaluation relies entirely on generic, holistic image metrics (SSIM, LPIPS, FID) which are not specifically sensitive to high-frequency content.

5. Unprofessional Formulation and Unexplained Notation: 1. The notation in formulas is highly inconsistent, mixing symbols like *, ·, and \times arbitrarily. 2. The term "Overall Objective Loss" is non-standard and awkward. 3. Numerous symbols are defined once and never used again (e.g., A(i) \ P(i)), cluttering the text without improving understanding.

6. Inadequate User Study Design: The user study is critically under-explained. It is unclear if all 20 volunteers evaluated the same set of 100 samples or if the samples were randomized per user. In addition, with only 20 participants, the sample size is too small to draw statistically significant conclusions.

**Questions:**

see weakness.

---

> ### Author Response · Authors · 2025-11-21
>
> Thank you for the constructive reviews. We will address your concerns below. In our revised version, changes are all highlighted by blue colored texts.
>
> ## Related to Weaknesses
>
> ### To W1:
> Thanks for this insightful comment. In the revised version, **we have added comparisons with the most recent state-of-the-art methods from 2025, including DA-Font (ACM MM 2025) and MX-Font++ (ICASSP 2025)**, presented in Section 4.3 (Page 8) and Section A.3 (Page 14). These additional results further demonstrate that HF-Font achieves superior performance in terms of both local detail preservation and overall style fidelity. Besides, **we also added cross-lingual font generation experiments** (Section A.7 on Page 18) to further demonstrate the robustness and generalization ability of our method.
>
> ### To W2:
> Thanks for this suggestion. However, HF-Font focuses on font generation, and **the standard practice protocol in this field, as adopted by MX-Font (ICCV 2021), CF-Font (CVPR 2023), DA-Font (ACM MM 2025), and other representative works, is based on individual glyphs rather than complete sentences.** Sentence-level generation belongs to the text rendering task, which involves additional factors such as spacing and layout, and is therefore beyond the scope of font style generation. Consequently, **generating full sentences is not applicable to our task.** Moreover, our qualitative evaluations follow the established practices and already cover a sufficiently diverse set of characters to ensure fairness and completeness.
>
> ### To W3:
> Thanks for the comment. To be carefully noted, the visual generation results in Figures 6, 12, and 15 clearly show that **our HF-Font produces sharper stroke boundaries and more accurate local structures, resulting in outputs that are noticeably closer to the Ground Truth**. By contrast, the methods compared in Section 4.3 (Page 8) and Section A.3 (Page 14) do not utilize high-frequency features, which contributes to their less satisfactory visual results. These improvements directly stem from the effective use of high-frequency style features. Furthermore, **the ablation study in Section 4.4 (Page 9) confirms that removing high-frequency components leads to noticeable performance degradation**. Overall, these above results demonstrate that high-frequency information is effectively exploited in our framework.
>
> ### To W4:
> Thanks for the assessment. We sincerely appreciate your suggestion regarding the high-frequency-specific quantitative metrics. Nevertheless, it is worth noticing that **in font generation research, the widely-adopted quantitative evaluation metrics such as SSIM, LPIPS, and FID are standard and provide reliable holistic assessment** at both pixel and perception levels. While these metrics are not explicitly high-frequency-specific, the effectiveness of high-frequency features in HF-Font is clearly demonstrated through comparison experiments (Section 4.3, Page 8; Section A.3, Page 14), ablation studies (Section 4.4, Page 9), extensive generation results (Section A.6, Page 17), and cross-lingual font generation results (Section A.7 on Page 18). These analyses together demonstrate that incorporating high-frequency information significantly improves stroke clarity, local structures, and overall resemblance to the Ground Truth.
>
> ### To W5:
> Thanks for the suggestion. The notations in our formulas are consistent throughout the whole manuscript, and all symbols are defined at their first occurrence to ensure clarity and facilitate understanding of the proposed method. Besides, in our revision, **we have replaced "Overall Objective Loss" with a more standard term "Total Loss"** (line 345, Page 7). Furthermore, symbols such as $A(i) \backslash P(i)$ are included to clarify the derivation and computation of the formulations, supporting a complete and rigorous exposition.
>
> ### To W6:
> Thanks for the comment. In our experiments, **all the 20 volunteers evaluated the same set of 100 samples** to ensure consistency and comparability across participants. In the revised version, **we have added a clarification regarding this** in lines 371-372 (Page 7). Besides, although the number of participants is not large, it is sufficient to provide indicative insights on perceptual quality for the purpose of our research. The user study results, combined with quantitative and qualitative comparison results presented in the Section 4.3 (Page 8) and Section A.3 (Page 14), collectively support the effectiveness of HF-Font.
>
> Once again, we gratefully thank you for your time and hope to meet your expectations.

---

### Official Review · Reviewer_Diqt · 2025-11-01

**Soundness:** 3
**Presentation:** 3
**Contribution:** 3
**Rating:** 4
**Confidence:** 4

**Summary:**

This paper focuses on few-shot font generation, which creates new fonts with a limited number of glyph references. The current models are still hard to capture delicate glyph details, thus resulting in stroke errors, artifacts, and blurriness. Authors develop a novel style-enhanced module to improve the style extraction by incorporating high-frequency features from reference images. Authors also develop a Style-Content Fusion Module (SCFM) to integrate style features with a component-wise codebook for encoding content semantics.

**Strengths:**

To handle stroke errors, artifacts, and blurriness in FFG task, the authors develop a novel style-enhanced module to improve the style extraction and a Style-Content Fusion Module (SCFM) to integrate style features. A style contrastive loss is developed to better transfer high-frequency features. The paper is well-written and easy to follow.

**Weaknesses:**

1. About the insight. In Fig.1, we can find that there are some texture patterns (in white lines) in high-frequency images. However, I think these patterns may have no contribution to the font style.
2. Since the authors claim high-frequency information is important for style features, the authors would be better to provide visualization results to support this claim.

**Questions:**

1. As I know, the font generation task aims to generate images with two colors, black and white. How to define the high-frequency in this case?
2. I think the style information can be separated into two parts: global and local. The high-frequency information maybe only affect the local style details.
3. The authors claim "further enlarging the codebook yields marginal gains, with some metrics (e.g., RMSE) even showing slight degradation. This suggests that while a larger codebook could enhance representation, excessive sizes would introduce more redundant information, which might compromise the model’s efficiency". I notice the codebook size is also an important hyper-parameter in other tasks, such as VAE. Do the experimental results have similar trends?

---

> ### Author Response · Authors · 2025-11-21
>
> Thank you for the helpful reviews. We will address your concerns below. In our revised version, changes are all highlighted by blue colored texts.
>
> ## Related to Weaknesses
>
> ### To W1:
> The white texture-like patterns in the high-frequency images are not irrelevant noise. Instead, **they reflect local glyph structures,** such as curvature transitions, stroke junction variations, and edge intensity changes. These components provide useful cues for **maintaining sharpness and reducing blurring or artifacts** in the generated results. In our HF-Font, **both the Style-Content Fusion Module (SCFM) and the style contrastive loss** explicitly utilize these high-frequency signals. Besides, our qualitative comparisons and generation results (Figures 5, 10, 14, 15, and 16) consistently show that **incorporating high-frequency features** could lead to better local clarity and style fidelity. Additionally, in the revised manuscript, **we also included cross-lingual font generation experiments** (Section A.7 on Page 18) to further demonstrate the robustness and generalization ability of our method.
>
> ### To W2:
> Thanks for the suggestion. In fact, our paper already provides several visualizations that support the contribution of high-frequency information. Specifically, the improvements in **local sharpness, stroke boundaries, and artifact reduction** can be clearly observed in the generation results across diverse settings (e.g., Figures 5, 10, 14, 15, and 16). These examples show that incorporating high-frequency signals enables the model to **better preserve fine-grained structural details** compared with methods that do not leverage such features. In our revision, **we have added a clarification regarding this** in lines 428-429 on Page 8.
>
> ## Related to Questions
>
> ### To Q1:
> In font images, high-frequency is defined by the **spatial rate of intensity change** rather than the number of colors. Although glyph images are visually black-and-white, **their stroke boundaries still contain sharp gradients** caused by curvature variations, junction structures, and so on. These regions produce clear high-frequency responses when applying a high-pass filter. Therefore, in our setting, the high-frequency components refer to these **stroke-edge transitions and fine structural variations** instead of color differences. In our revision, **we have clarified this point** in lines 53, 68-70 on Pages 1-2.
>
> ### To Q2:
> Thanks for the concern. We agree that high-frequency signals primarily capture local style variations. To ensure **global style consistency**, our HF-Font employs an alignment mechanism, **the "Content Alignment Module (CAM)",** which attends to reference glyphs with similar content elements to guide the style transfer process. This design allows the model to integrate local details while **maintaining coherent global structural patterns**, thus achieving fine-grained local fidelity without compromising overall stylistic consistency.
>
> ### To Q3:
> Thanks for the observation. In our work, we systematically evaluated codebook sizes of 50, 75, 100, 125, and 150 for HF-Font in Table 3 (Page 9), and **found that 100 provides the best trade-off** between the representation capacity and efficiency. While we acknowledge that codebook size is also an important hyper-parameter in other tasks such as VAE, **the optimal size and trends are task-dependent.** Differences in data distribution, feature representations, and objectives mean that enlarging the codebook may have different effects in other settings. Therefore, our analysis is specific to the few-shot font generation task, and similar trends may not necessarily hold in other settings.
>
> Once again, we gratefully thank you for your time and hope to meet your expectations.

---

### Meta-Review · Area_Chair_TAYx · 2025-12-17

**Summary:**

3/4 reviewers unanimously give a clear rejection. Two common problems are the concept similarity with the prior works and the contribution of high-frequency enhancement. After reading the discussion, I stand with the reviewers and hold that the concerns are not well addressed, so I am inclined to give a rejection.

**Reviewer Concerns:**

There are some important concerns that are not well addressed
1. Reviewer B4R3 and nBGk both presented an important question on the difference with the previous works like One-DM and DA-Font while the authors' rebuttal is rather weak.
2. The positive role of high-frequency enhancement is not clearly presented, for example, the difference performance of different high-pass filters , which is questioned by all the reviewers.

**Reviewer Scores:**

I believe that Reviewer fLfM, B4R3 and nBGk, the latter two reviewers in particular, would not be satisfied with the rebuttal since the authors haven't shown concrete results to prove their claims.

---

### Decision · Program_Chairs · 2026-01-26

Reject